# Bi-allelic *MCM10* variants associated with immune dysfunction and cardiomyopathy cause telomere shortening

Ryan M. Baxley [1], Wendy Leung[1,11], Megan M. Schmit [1,11], Jacob Peter Matson [2,11], Lulu Yin[1], Marissa K. Oram [1], Liangjun Wang[1], John Taylor[3], Jack Hedberg[1], Colette B. Rogers[1], Adam J. Harvey[1], Debashree Basu[1], Jenny C. Taylor[4,5], Alistair T. Pagnamenta [4,5], Helene Dreau[6], Jude Craft[7], Elizabeth Ormondroyd [8], Hugh Watkins [8], Eric A. Hendrickson [1], Emily M. Mace [9], Jordan S. Orange[9], Hideki Aihara [1], Grant S. Stewart [10], Edward Blair[7], Jeanette Gowen Cook [2] & Anja-Katrin Bielinsky [1✉]

Minichromosome maintenance protein 10 (MCM10) is essential for eukaryotic DNA replication. Here, we describe compound heterozygous *MCM10* variants in patients with distinctive, but overlapping, clinical phenotypes: natural killer (NK) cell deficiency (NKD) and restrictive cardiomyopathy (RCM) with hypoplasia of the spleen and thymus. To understand the mechanism of MCM10-associated disease, we modeled these variants in human cell lines. MCM10 deficiency causes chronic replication stress that reduces cell viability due to increased genomic instability and telomere erosion. Our data suggest that loss of MCM10 function constrains telomerase activity by accumulating abnormal replication fork structures enriched with single-stranded DNA. Terminally-arrested replication forks in MCM10-deficient cells require endonucleolytic processing by MUS81, as *MCM10:MUS81* double mutants display decreased viability and accelerated telomere shortening. We propose that these bi-allelic variants in *MCM10* predispose specific cardiac and immune cell lineages to prematurely arrest during differentiation, causing the clinical phenotypes observed in both NKD and RCM patients.

[1] Department of Biochemistry, Molecular Biology, and Biophysics, University of Minnesota, Minneapolis, MN 55455, USA. [2] Department of Biochemistry and Biophysics, University of North Carolina, Chapel Hill, NC 27599, USA. [3] Oxford Medical Genetics Laboratories, Oxford University Hospitals NHS Foundation Trust, Oxford, UK. [4] Wellcome Centre Human Genetics, University of Oxford, Oxford OX3 7BN, UK. [5] Oxford NIHR Biomedical Research Centre, Oxford OX3 7BN, UK. [6] Department of Haematology, University of Oxford, Oxford OX3 7BN, UK. [7] Oxford Centre for Genomic Medicine, Oxford University Hospitals NHS Foundation Trust, Oxford, UK. [8] Division of Cardiovascular Medicine, Radcliffe Department of Medicine, University of Oxford, Oxford, UK. [9] Vagelos College of Physicians and Surgeons, Columbia University, New York, NY 10032, USA. [10] Institute of Cancer and Genomic Sciences, University of Birmingham, Birmingham B15 2TT, UK. [11] These authors contributed equally: Wendy Leung, Megan M. Schmit, Jacob Peter Matson. ✉email: bieli003@umn.edu

The DNA replication program has evolved to ensure the fidelity of genome duplication. Conceptually, this program can be divided into replication origin licensing, origin firing, and DNA synthesis. Origins are licensed by loading double hexamers of the core replicative helicase composed of minichromosome maintenance complex proteins 2 to 7 (MCM2-7)[1,2] onto DNA. Origin firing requires helicase co-activators, including the *go-ichi-ni-san* (GINS) complex and cell division cycle protein 45 (CDC45), to form the CDC45:MCM2-7:GINS (CMG) helicase[3,4]. Upon firing, minichromosome maintenance protein 10 (MCM10) is essential for the CMG complexes to reconfigure and bypass each other as double-stranded DNA is unwound bi-directionally[5–7]. In addition, as DNA synthesis proceeds, MCM10 stabilizes the replisome to prevent replication stress and promote genome stability[8–10].

During oncogenic transformation, cells overexpress replication factors to drive proliferation[7,11]. Therefore, it is not surprising that *MCM10* is commonly upregulated in cancer cell lines and tumor samples, suggesting that transformed cells rely on MCM10 to prevent genome instability from reaching lethal levels[7]. For many years these observations constituted the only examples of *MCM10*-associated human disease. Recently, we identified human germline *MCM10* variants in two unrelated families that were associated with distinct phenotypes: natural killer (NK) cell deficiency (NKD)[12] and restrictive cardiomyopathy (RCM) associated with thymic and splenic hypoplasia. Both of these conditions appear to be the result of compound heterozygous *MCM10* variants. Previous studies of human *MCM10* have relied heavily on overexpression of epitope-tagged constructs and/or transient knockdown, and thus poorly recapitulated the effects of these variants[7]. To gain a more accurate understanding of the cellular phenotypes underlying *MCM10*-associated NKD or RCM, we chose to model these variants in human cell lines.

Here, we demonstrate that human *MCM10* is haploinsufficient in transformed HCT116 and non-transformed telomerase immortalized hTERT RPE-1 cells (referred to subsequently as RPE-1). These phenotypes were more severe in HCT116 cells, disrupting normal cell cycle distribution and affecting global DNA synthesis due to decreased origin firing. Chronic MCM10 deficiency in both cell types caused increased cell death and revealed an unexpected telomere maintenance defect. Our data suggest that telomeric replication fork stalling compromised telomerase access to chromosome ends. Telomere erosion was not a characteristic of *CDC45* or *MCM4* haploinsufficiency, indicating that the telomeric replisome is uniquely reliant on robust MCM10 levels. Finally, our data demonstrate that in *MCM10:MUS81* double mutants, the growth and telomere maintenance phenotypes were exacerbated. Although likely not the only nuclease involved, our data demonstrate that *MCM10* mutants utilize MUS81 to process stalled replication forks and improve cell survival. Taken together, our results revealed that MCM10 is critical for human telomere replication and suggest that defective telomere maintenance caused both *MCM10*-associated NKD and RCM.

## Results

**Modeling *MCM10* patient variants reveals haploinsufficiency in human cell lines**. Compound heterozygous *MCM10* variants were identified in unrelated patients that presented with NKD or fetal RCM with thymic and splenic hypoplasia (Table 1; all genetic notation is in reference to *MCM10* transcript NM_018518.5). Clinical and genetic analysis of the NKD patient and family was recently described[12]. The NKD-associated variants were identified in a single-family and included one missense (c.1276 C > T, p.R426C) and one nonsense variant (c.1744 C > T, p.R582X). Clinical and genetic analysis of the RCM patients and

family are described in Supplementary Note 1. The RCM-associated variants were identified in three affected siblings and included one splice donor site (c.764 + 5 G > A, p.D198GfsTer10) and one frameshift variant (c.236delG, p.G79EfsTer6; Fig. 1a, b, Supplementary Fig. 1a, b). Homozygous individuals were not reported in the Genome Aggregation Database (gnomAD)[13] for any of the RCM-associated *MCM10* variants. Furthermore, allele frequencies reported for these variants are consistent with very rare, autosomal recessive alleles (Table 1). Both the NKD-associated missense and RCM-associated splice site variants map to the conserved MCM10 internal domain (ID; Supplementary Fig. 1d). The NKD-associated p.R426C substitution resides C-terminal to zinc-finger 1 and was not predicted to significantly alter protein structure (Fig. 1c). Consistently, RPE-1 cells homozygous for this variant showed stable MCM10 expression, although analyses of proliferation and sensitivity to ultra-violet (UV) light demonstrated that this allele is hypomorphic[12].

The RCM-associated c.764 + 5 G > A variant altered the exon 6 splice donor sequence. This variant is predicted to decrease efficiency of the splice donor by approximately half, but would not cause complete inactivation (MaxEntScan[14]; Supplementary Fig. 1b). In fact, the exon 6 variant splice donor retains a higher functional score than the wild-type splice donor for *MCM10* exon 5. Therefore, this variant allele likely produces a reduced amount of wild-type *MCM10* transcript. RNA analysis suggested that this variant also allowed splicing of exon 5 to exon 7, excluding exon 6. As a consequence, the open reading frame changed and introduced a premature stop codon at position c.791-793, leading to degradation of the transcript or expression of a protein truncated at amino acid 198 (r.593_764del; p.D198GfsTer10; Supplementary Fig. 1b, d). This protein, if it were expressed, would be truncated prior to the nuclear localization sequence and remain cytoplasmic[15]. Notably, our analysis also identified a cryptic splice acceptor in exon 7 with a reduced functional score in comparison to the canonical exon 7 acceptor, although higher than the exon 5 acceptor (MaxEntScan[14]; Supplementary Fig. 1b). Use of this cryptic site could lead to expression of MCM10 carrying an internal deletion of amino acids 198-262 (Supplementary Fig. 1d). The first 38 amino acids of this region are part of the unstructured linker domain that connects the N-terminus with the ID. The adjacent 27 amino acids are part of the OB-fold, the major MCM10 DNA binding domain, and would delete half of the conserved MCM10 alpha-helix 1 (Fig. 1c)[16]. To gain insight into how this internal deletion might affect MCM10 function, we compared the biochemical behaviors of the hMCM10-ID (residues 236-435) and a corresponding N-terminally truncated mutant spanning amino acids 263-435 (ΔN27; Fig. 1c). Size-exclusion chromatography showed that, although both proteins were predominantly monomeric, ΔN27 was partially aggregated and eluted in the void volume (Fig. 1d). In addition, to assess protein secondary structure we recorded circular dichroism (CD) spectra at 25 °C for the wild type and ΔN27 ID. These spectra showed pronounced negative ellipticity peaks at 205 nm, which is typical for a zinc finger protein[17] (Supplementary Fig. 1e) and suggested that these proteins had similar secondary structures. The spectrum of the wild type protein at 60 °C remained mostly unchanged, indicating a stable protein structure. However, the spectrum of the ΔN27 mutant had a significantly reduced 205 nm peak at 60 °C, indicative of decreased thermal stability (Supplementary Fig. 1e). Next, we collected thermal denaturation data by monitoring CD signals at 205 nm in a temperature range of 40 to 90 °C. From the single wavelength curves the melting temperatures ($T_m$) of wild type hMCM10-ID and the mutant ΔN27 were determined to be 71 °C and 62 °C, respectively (Supplementary Fig. 1f). Overall, these experiments demonstrated that the N-terminal deletion leads to

**Table 1 MCM10 patient variants associated with NKD and fetal RCM.**

| Variant | Mutation Type | Location | Functional Class | gnomAD frequency | Pathology |
|---|---|---|---|---|---|
| c.236delG, p.G79EfsTer6 | Frameshift | Exon 3 | LOF | n.r. | Restrictive cardiomyopathy with lymphoreticular hypoplasia |
| c.764 + 5 G > A, r.593_763del, p.D198GfsTer10 | Splice Donor | Intron 6 | hypomorph | 1/232,426 | |
| c.1276 C > T, p.R426C | Missense | Exon 10 | hypomorph | 7/282,582 | NK cell deficiency |
| c.1744C > T, p.R582X | Nonsense | Exon 13 | LOF | n.r. | |

Variants annotated in reference to MCM10 transcript NM_018518.5.
Frequencies based on gnomADv.2.1.1[13]. LOF = loss of function, n.r. = not reported.

compromised stability of hMCM10-ID and makes the protein prone to aggregation. Taken together, these data suggest that like the NKD-associated missense allele, the *MCM10* splice donor variant is a hypomorphic allele but causes a more severe reduction in MCM10 function.

Importantly, both hypomorphic variants caused pathological defects in patients only in combination with a second *MCM10* variant that introduced a premature stop codon. To understand the impact of these variants on MCM10 function, we modeled them in HCT116 cells, a cell line amenable to multiple gene targeting methodologies. Initially, we utilized recombinant adeno-associated virus (rAAV) to target exon 14 and cause a frameshift that introduced a premature stop codon resulting in a p.F583X mutation to model the NKD-associated p.R582X variant (Fig. 1a, Supplementary Fig. 1d). Sanger sequencing of the remaining functional allele confirmed that exon 14 retained wild type coding and splicing sequences (Supplementary Fig. 2a). Next, clustered regularly interspersed palindromic repeat-CRISPR-associated 9 (CRISPR-Cas9) was used to introduce frameshift mutations causing premature stop codons in exon 3 to model the RCM-associated variant in the same exon (Fig. 1a, Supplementary Fig 1c, Supplementary Fig 2b, c). Due to the introduction of premature stop codons, these variants presumably led to nonsense-mediated mRNA decay. Regardless, if translated, MCM10 would be truncated prior to the nuclear localization sequence and remain cytoplasmic[15]. Western blot analysis demonstrated stable Mcm10 reduction in both exon 14 and exon 3 targeted *MCM10*[+/-] cell lines (Fig. 1e), which significantly slowed cell proliferation (Fig. 1f) and decreased clonogenic survival (Fig. 1g, h), due in part to increased cell death (Fig. 1i, Supplementary Fig. 2d). These observations demonstrated that — in HCT116 cells — *MCM10* is genetically haploinsufficient.

**Reduced origin firing causes DNA replication defects and impairs viability of Mcm10-deficient cells**. To define the cause of the growth defect in *MCM10*[+/-] mutants, we utilized quantitative analytical flow cytometry. Cell cycle distribution was assessed by staining with 4′,6-diamidino-2-phenylindole (DAPI) for DNA content in combination with a pulse of 5-ethynyl-2′-deoxyuridine (EdU) to label S-phase cells (Fig. 2a). In *MCM10*[+/-] cells, we detected a significant increase in G1- and a decrease in S-phase populations (Fig. 2b) indicative of delayed progression through the G1/S-phase transition. Next, we measured chromatin-bound MCM2 and EdU-labeled DNA to quantify G1-phase origin licensing and S-phase DNA synthesis (Fig. 2c-d)[18]. Origin licensing in *MCM10*[+/-] mutants was identical to wild type cells (Fig. 2e). However, we observed a significant DNA synthesis defect during the 30-minute labeling pulse in *MCM10*[+/-] cells (Fig. 2f). These data demonstrated a requirement for MCM10 in DNA synthesis, but not origin licensing, and are consistent with the published roles for Mcm10 in DNA replication[7].

To understand the DNA synthesis defect in *MCM10*[+/-] cells, we performed DNA combing. We first measured inter-origin distance (IOD) to determine if origin firing was perturbed. Whereas wild-type cells displayed an average IOD of ~91 kb, the IOD in *MCM10*[+/-] cells markedly increased to ~120 kb (Fig. 2g). This difference equates to ~25% fewer origin firing events in *MCM10* mutants. Next, we measured global fork speed and stability. Fork speed was modestly, but significantly, increased in *MCM10*[+/-] cells (Fig. 2h), consistent with the inverse regulation of fork speed and origin firing in eukaryotes[19,20]. Fork stability was not changed (Fig. 2i). These data suggested that MCM10 deficiency reduced the number of active forks. To confirm this idea, we assessed ubiquitination (Ub) of proliferating cell nuclear antigen (PCNA), because this modification at lysine 164 occurs specifically at active forks in response to replication stress[21]. Since HCT116 cells exhibit intrinsic replication stress, PCNA-Ub is detectable even under unperturbed conditions (Fig. 2j)[22]. To enhance PCNA-Ub, we exposed cells to 40 J/m[2] of UV radiation. As expected, PCNA-Ub was 1.5- to 3-fold higher in wild-type cells than in *MCM10*[+/-] cells (Fig. 2j). Furthermore, phosphorylated RPA32 was elevated in wild-type cells in comparison to *MCM10*[+/-] cells after exposure to UV radiation. Therefore, MCM10 deficiency decreased origin firing and resulted in fewer active forks.

**Replication stress in Mcm10-deficient cells causes genomic instability and spontaneous reversion of the *MCM10* mutation**. To understand if chronic MCM10 deficiency promoted genome instability, we generated late passage HCT116 wild type and *MCM10*[+/-] populations. As *MCM10*[+/-] cells were passaged, we observed an increase in cells with abnormal morphology and multiple enlarged or pyknotic nuclei indicative of decreased viability (Supplementary Fig. 3a). Karyotype analysis of wild-type cells after ~200 population doublings (PDs) as well as *MCM10*[+/-] populations after mid- (~25 PDs) and late passage (~100 PDs) revealed increased genome instability in the mutants (Supplementary Fig. 3b). Late passage wild-type cells harbored three populations represented by the parental HCT116 karyotype and two derivatives, with 10% of karyotypes carrying unique aberrations. Mid-passage *MCM10*[+/-] cells were comprised of one major population distinct from the parental karyotype carrying two novel aberrations, as well as seven additional novel karyotypes. Overall, 23% of mid-passage *MCM10*[+/-] karyotypes had unique aberrations. Furthermore, late passage *MCM10*[+/-] cells consisted of a clonal population distinct from the parental karyotype carrying three novel aberrations, as well as nineteen additional novel karyotypes. Overall, 63% of late-passage *MCM10*[+/-] karyotypes had unique aberrations, suggesting that they were the result of independent events. The acquired chromosomal rearrangements in *MCM10* mutants significantly overlapped with common fragile sites (CFSs) (80%; Supplementary Table 1)[23], which is consistent with a role for MCM10 in preventing fragile site breakage[24].

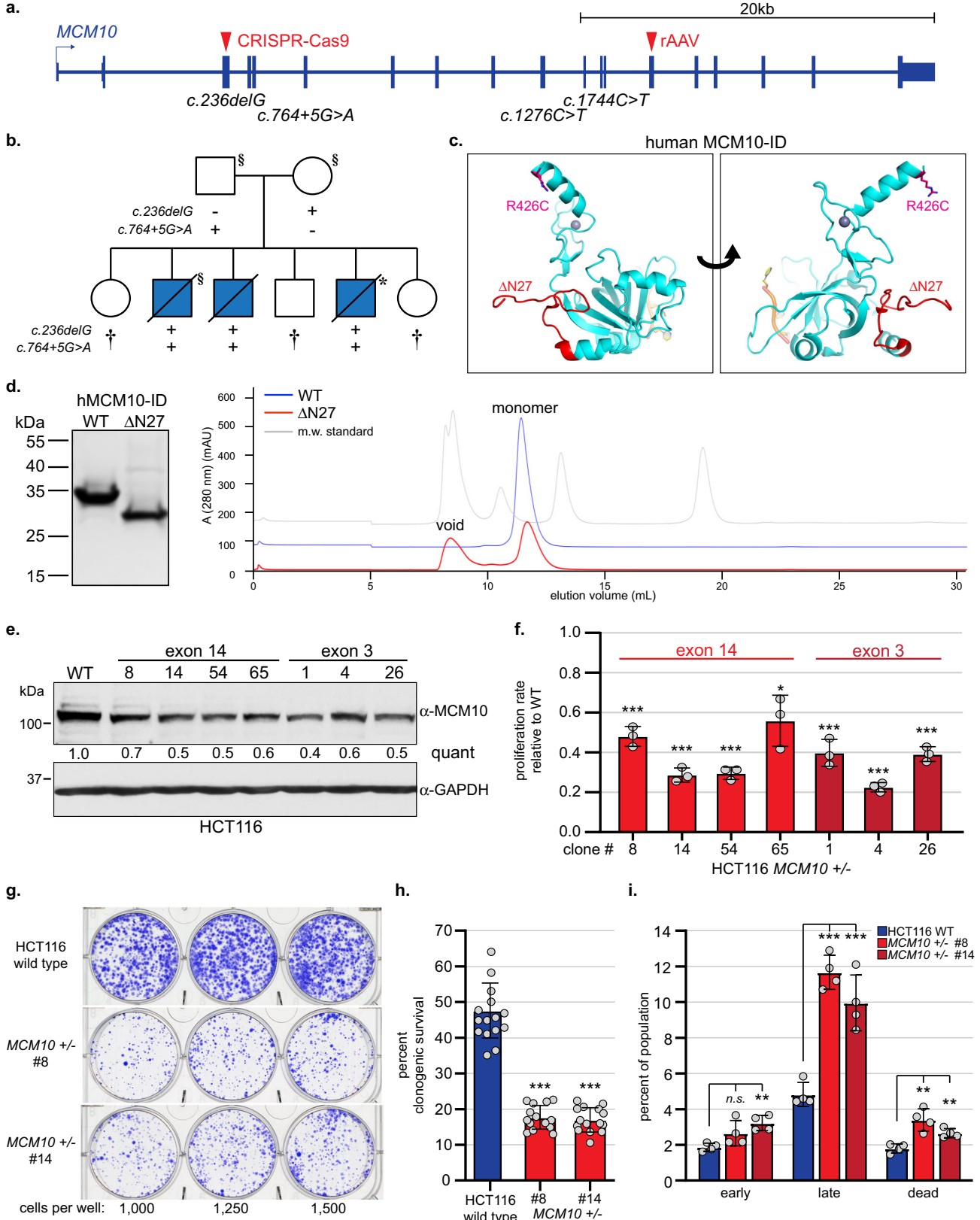

The overlap of chromosomal aberration hot spots with CFSs prompted us to investigate telomere maintenance, as telomeres are also origin-poor and hard-to-replicate regions[25]. We performed telomere restriction fragment (TRF) length analysis to measure average length over time. Whereas, wild-type telomeres were stable, telomeres in *MCM10*[+/-] cell lines were shorter at early passage and eroded over time (Fig. 3a). We confirmed this phenotype in early passage exon 3 *MCM10*[+/-] HCT116 cells, which showed significantly eroded telomeres (Supplementary Fig. 3c). Subsequent analyses focused on exon 14 *MCM10* mutant cell lines. To independently evaluate telomere shortening, we performed telomere fluorescence in situ

**Fig. 1 Modeling *MCM10* patient-associated variants. a** *MCM10* schematic indicating NKD- (c.1276 C > T;c.1744C > T) and RCM-associated (c.236delG; c.764 + 5 G > A) variants and exons targeted using CRISPR-Cas9 (exon 3) or rAAV (exon14). **b** Pedigree and segregation of *MCM10* variants in the RCM family. Blue shading indicates fetal RCM. Individuals that underwent exome (§) or genome (*) sequencing are indicated. Clinically unaffected children that are not carriers of both pathogenic variants (†) are indicated (carrier status of minors not disclosed). **c** Model of hMCM10-ID bound to single-stranded DNA, based on *Xenopus laevis* MCM10-ID (Protein Data Bank codes 3EBE[16] [https://www.ncbi.nlm.nih.gov/Structure/pdb/3EBE] and 3H15[74] [https://www.ncbi.nlm.nih.gov/Structure/pdb/3H15]). Locations of the R426C (pink) and ΔN27 (red) variants and the zinc ion (gray sphere) are shown. **d** (Left) Coomassie blue-stained gel of WT and ΔN27 hMCM10-ID. (Right) Chromatography profiles of WT and ΔN27 hMCM10-ID. The molecular weight standard (gray) included thyroglobulin (670 kDa), γ-globulin (158 kDa), ovalbumin (44 kDa), myoglobin (17 kDa), and vitamin B12 (1.3 kDa). Trace profiles for WT and the molecular weight standard are offset on the y-axis for display purposes. **e** Western blot for MCM10 with GAPDH as a loading control. Quantification of MCM10 levels normalized to loading control, relative to wild type is indicated. **f** Average proliferation rate in *MCM10*$^{+/-}$ cells normalized to wild type. For each cell line $n = 6$ replicates, with average values for biological replicates indicated (gray circles). **g** Comparison of clonogenic survival of HCT116 wild type (top) and *MCM10*$^{+/-}$ cells (middle/bottom). Cells plated per well are noted. **h** Average percentage clonogenic survival in HCT116 wild type (blue) and *MCM10*$^{+/-}$ cells (red), $n = 15$ replicates. Individual data points are indicated (gray circles). **i** Average percentage of early apoptotic, late apoptotic, or dead cells. Wild type (blue) and clonal *MCM10*$^{+/-}$ cell lines (red) are shown with $n = 4$ biological replicates. Individual data points are indicated (gray circles). Error is indicated in **f**, **h**, and **i** as SD and significance was calculated using an unpaired, two-tailed student's *t*-test with *<0.05; **<0.01, ***<0.001. Source data for panels **d**, **e**, **f**, **h**, and **i**, including relevant exact p-values, are provided in the Source Data file.

hybridization (t-FISH) of metaphase chromosomes. Consistent with our TRF analyses we observed a significant increase in chromosomes lacking a telomere signal ("signal-free ends") in *MCM10*$^{+/-}$ metaphase spreads (Fig. 3b). We also quantified fragile telomeres, chromosome ends with multiple telomere foci that are indicative of abnormal structures[26], but did not find any increase in *MCM10*$^{+/-}$ mutants (Fig. 3b). Next, we measured β-galactosidase (β-gal) activity to determine whether *MCM10*$^{+/-}$ cells activated cellular senescence pathways[27,28]. We observed significantly higher activity in mutant cell extracts, further corroborating the telomere maintenance defect, although this does not rule out other sources of stress-induced senescence pathway activation[29] (Supplementary Fig. 3d). Finally, we measured telomerase activity using the telomeric repeat amplification protocol (TRAP)[30], which ensured that activity was equivalent in all cell lines (Fig. 3c). We could thus exclude telomerase inactivation as the cause of telomere erosion. Our data indicate that the telomere maintenance defect in *MCM10* mutant cells is one feature of the underlying defect in maintaining genome stability.

To understand whether MCM10 deficiency might drive cells into telomere crisis, we propagated six independent *MCM10*$^{+/-}$ populations for >75 PDs. TRF analyses documented that these populations contained telomeres 2 to 4 kb in length (Fig. 3d). Consistent with these data, the frequency of signal-free ends in two additional independent late passage *MCM10*$^{+/-}$ populations remained elevated in *MCM10*$^{+/-}$ cells (Fig. 3b, Supplementary Fig. 3e). Again, we did not detect changes in telomere fragility (Supplementary Fig. 3e). Thus, although TRF analyses indicated that telomeres in *MCM10* mutants eroded over time, there was insufficient erosion to detect a quantifiable increase in signal-free ends on metaphase chromosomes. This observation suggested that late passage *MCM10*$^{+/-}$ populations maintained minimal telomeres in order to remain viable. To evaluate this idea, we first confirmed the genotype of each population. Both late passage populations analyzed by t-FISH showed the expected mutant genotype (Supplementary Fig. 3f). Unexpectedly, only three populations carried the heterozygous PCR pattern and three populations had spontaneously reverted the exon 14 mutation (Fig. 3e, f). Reversion resulted in two alleles carrying wild type coding sequence. Some cell lines retained the 3′ *loxP* site on both alleles, while others lost the *loxP* sites completely (Fig. 3e, f). These results prompted us to conduct additional long-term experiments closely monitoring PDs, telomere length, and genotype. Analysis of these time courses revealed two novel spontaneous reversion events (Fig. 3g). Importantly, *MCM10* reversion corresponded with rescued telomere length (Fig. 3g),

increased MCM10 levels (Fig. 3h), growth rate recovery (Fig. 3i), and rescued defects in cell cycle distribution (Fig. 3j).

To confirm that reversion events were not the misinterpretation of culture contamination with wild-type cells, we marked five independent *MCM10*$^{+/-}$ populations with a puromycin (PURO) resistance gene. Following drug selection, the genotype of each population was confirmed as heterozygous and cell lines were propagated and collected at regular time intervals. Each population underwent additional PURO selection twice during the time course (Supplementary Fig. 3g). After ~100 PDs, one of the five populations spontaneously reverted (Supplementary Fig. 3h). In this population, telomeres eroded initially but recovered and stabilized following reversion. We hypothesize that reversion occurred through a homology-dependent recombination mechanism (Supplementary Fig. 3i), such as break-induced replication (BIR)[31], following the random introduction of a DNA break at the *MCM10* locus, caused by the underlying increase in genomic instability in the mutant cell lines. Taken together, our data demonstrate that the toxic nature of MCM10 deficiency was actively selected against as *MCM10*$^{+/-}$ cells spontaneously reverted the locus to restore the *MCM10* coding sequence and rescue mutant phenotypes.

**Deficiency of essential replisome proteins CDC45 or MCM4 does not cause telomere erosion.** The mutant phenotypes in *MCM10*$^{+/-}$ cells led us to ask if HCT116 cells are broadly sensitive to heterozygosity of essential replisome genes. To test this, we constructed *CDC45*$^{+/-}$ and *MCM4*$^{+/-}$ cell lines. These genes encode CMG helicase proteins and contribute to the same processes that require MCM10[7]. *CDC45*$^{+/-}$ and *MCM4*$^{+/-}$ cell lines showed a reduction in the protein corresponding to the inactivated gene, but no change in MCM10 levels (Supplementary Fig. 4a). Unexpectedly, CDC45 levels were significantly reduced in *MCM4*$^{+/-}$ mutants, suggesting that MCM4 stabilizes CDC45 protein. We confirmed that CDC45 or MCM4 levels were not consistently reduced in *MCM10* mutants, implying that the mutant phenotypes in these cells are not attributable to reduced expression of either factor (Supplementary Fig. 4b). *CDC45*$^{+/-}$ and *MCM4*$^{+/-}$ populations proliferated slower than wild-type cells (Supplementary Fig. 4c), although this phenotype was not as severe as in *MCM10*$^{+/-}$ cells (Fig. 1f). The amount of chromatin-bound PCNA-Ub was similar in *CDC45*$^{+/-}$ and *MCM4*$^{+/-}$ mutants in comparison to wild-type cells, implying that the number of active forks was not reduced, unlike our observations in *MCM10*$^{+/-}$ cells (Fig. 2j, Supplementary Fig. 4d). Finally, TRF analyses revealed that telomere length was stable in *CDC45*$^{+/-}$

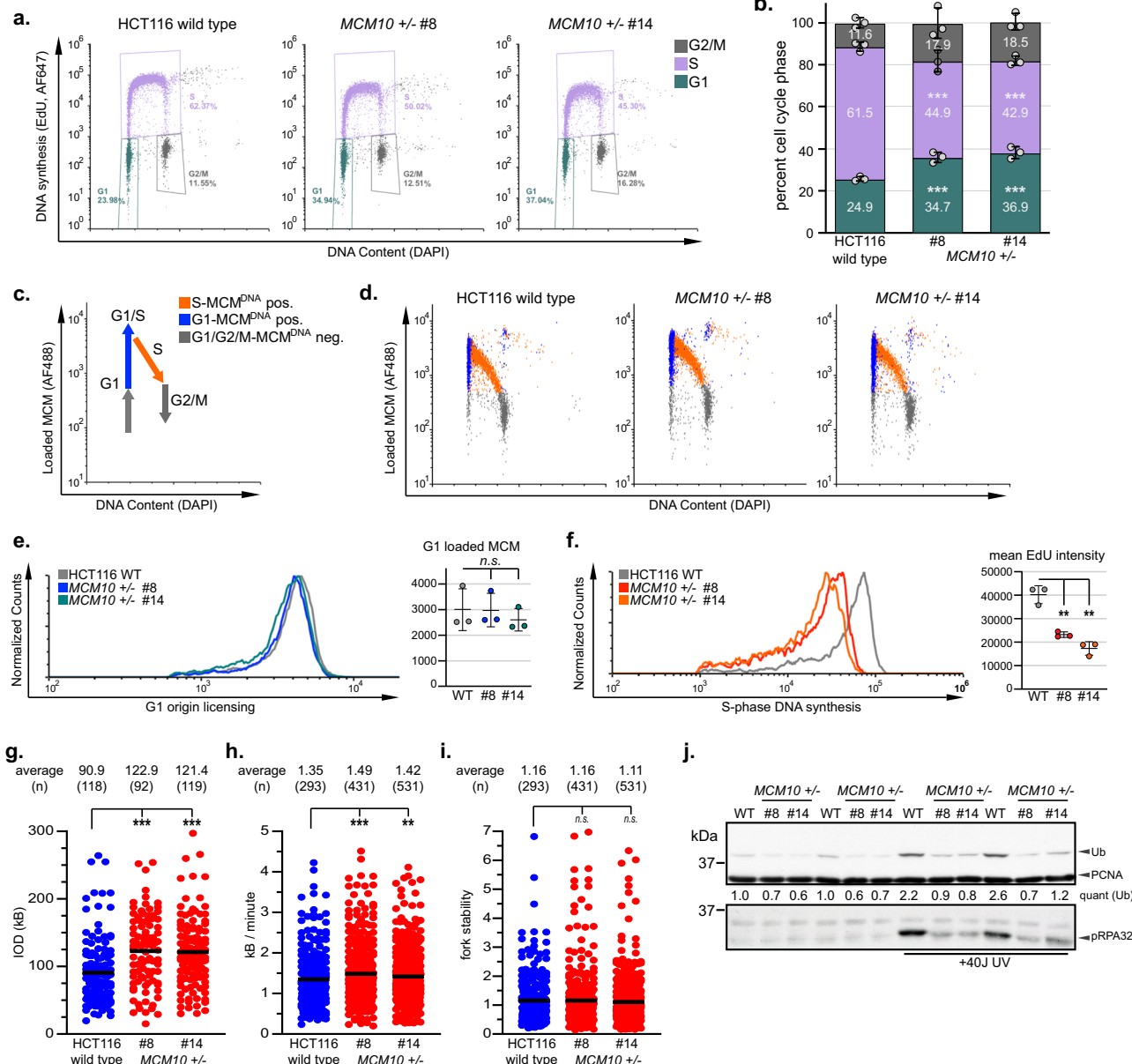

**Fig. 2 MCM10 deficiency causes significant cell cycle and DNA synthesis defects. a** Cell cycle distribution of wild type and $MCM10^{+/-}$ cells. The percentage of each population in G1- (green), S- (purple) or G2/M-phase (gray) is indicated. **b** Cell cycle distribution of wild type and $MCM10^{+/-}$ cell lines from three biological replicates. The average percentage of each population in G1- (green), S- (purple), and G2/M-phase (gray) is indicated. Individual data points are indicated (gray circles). **c** Schematic of flow cytometry analysis. Cell-cycle phase is defined by DNA content, EdU incorporation, and chromatin-loaded MCM2. **d** Analytical flow cytometry plots for wild type and $MCM10^{+/-}$ cells. G1-phase/MCM positive cells (blue), S-phase/MCM positive cells (orange) and G1- or G2/M-phase/MCM negative cells (gray) are indicated. **e** Comparison of origin licensing (left) and quantification of average G1 loaded MCM2 in biological replicates ($n = 3$) of wild type (gray) and $MCM10^{+/-}$ cells (blue/green). **f** Comparison of S-phase DNA synthesis (left) and average EdU intensity in biological replicates ($n = 3$) of wild type (gray) and $MCM10^{+/-}$ cells (red/orange). Error bars in **b**, **e**, and **f** indicate SD, and significance was calculated using an unpaired, two-tailed student's $t$-test with *<0.05; **<0.01, ***<0.001. **g** Inter-origin distance (IOD) quantification in wild type (blue) and $MCM10^{+/-}$ cells (red). Average IOD and number ($n$) quantified are listed. **h** Fork speed in wild type (blue) and $MCM10^{+/-}$ cells (red). Average fork speed (kb/minute) and number ($n$) quantified are listed. **i** Fork stability in wild type (blue) and $MCM10^{+/-}$ cells (red). Average fork stability and number ($n$) quantified are listed. Statistical significance for **g**–**i** was calculated using an unpaired, two-tailed Mann-Whitney Ranked Sum Test with *<0.05; **<0.01, ***<0.001. **j** Chromatin associated PCNA, PCNA-Ub, and phospho-RPA32, which binds to ssDNA exposed during replication stress, with and without 40 J UV treatment. Quantification of PCNA-Ub levels normalized to unmodified PCNA, relative to the first lane wild type sample is indicated. Source data for panels **b**, **e**, **f**, **g**, **h**, **i**, and **j**, including relevant exact p-values, are provided in the Source Data file.

and $MCM4^{+/-}$ mutants (Supplementary Fig. 4e, f). Taken together, these data argue that HCT116 cells are sensitive to inactivation of one $CDC45$ or $MCM4$ allele, but they are significantly more sensitive to $MCM10$ heterozygosity which is associated with a unique telomere maintenance defect.

**Modeling $MCM10$ variants in non-transformed RPE-1 cells confirms defects in telomere maintenance.** $MCM10$ haploinsufficiency was unexpected, as heterozygous mice were healthy and fertile[32] and haploinsufficiency is uncommon among human genes[33]. Moreover, the requirement for bi-allelic variants

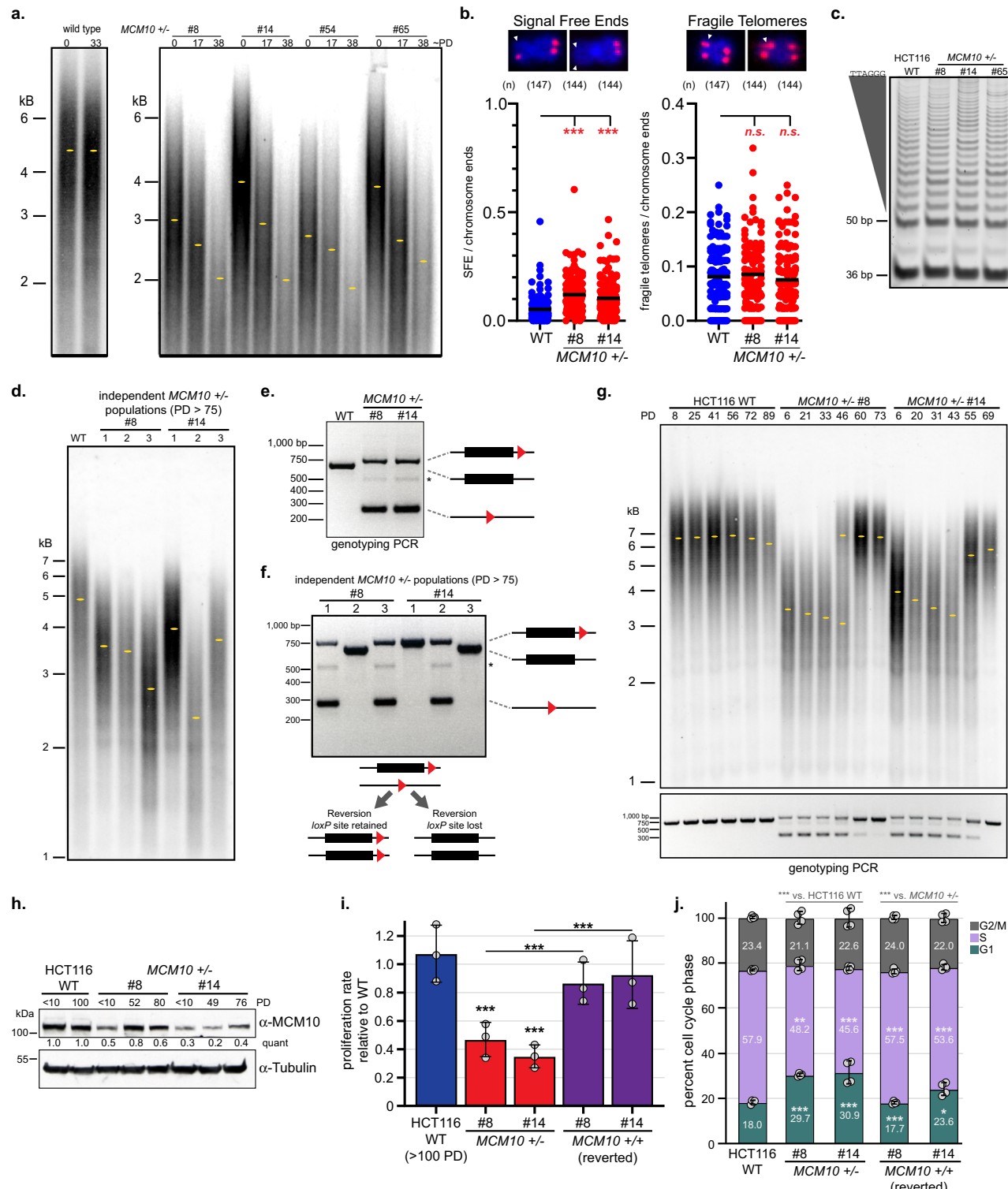

to elicit a clinical patient phenotype suggested that *MCM10* haploinsufficiency might be unique to human cell lines. To evaluate this idea, we modeled the *MCM10* exon 3 frameshift variant (c.236delG, p.G79EfsTer6) in RPE-1 cells. *MCM10*$^{+/-}$ RPE-1 cells showed stable reduction of MCM10 (Fig. 4a), which significantly slowed cell proliferation (Fig. 4b) and increased cell death (Fig. 4c). Significant changes in the cell cycle distribution of *MCM10*$^{+/-}$ cells were not observed (Fig. 4d, e) and the amount of origin licensing (Fig. 4f, g) and global EdU incorporation (Fig. 4h) appeared normal. However, these parameters would not detect

possible changes in origin activation. Therefore, we measured the levels of chromatin-bound PCNA-Ub as an indicator of the relative number of active replication forks. Although baseline PCNA-Ub is significantly lower than in HCT116 cancer cells, it is detectable under unperturbed conditions[34]. Because of this, we were able to determine that *MCM10* mutants had significantly lower PCNA-Ub (Fig. 4i), and thus fewer active replication forks. These findings recapitulate the origin firing defect and decrease in active forks observed in *MCM10* HCT116 mutants (Fig. 2g, j), albeit the phenotypes are less severe in RPE-1 cells. Notably,

**Fig. 3 MCM10 deficiency causes telomere erosion that is rescued following spontaneous reversion of the *MCM10* mutant locus. a** TRF analysis of HCT116 wild type and *MCM10*[+/−] cells. Estimated PDs and location of peak intensity (yellow ovals) are indicated. **b** Signal free-ends and fragile telomeres in wild type (blue) and *MCM10*[+/−] cells (red). Significance was calculated using an unpaired, two-tailed student's *t-test* with \*\*\*<.001; *n* = metaphases analyzed. Scale bars are 1 μm. **c** TRAP assay from wild type and *MCM10*[+/−] cells. The internal PCR control at 36 bp and telomerase products above 50 bp are noted. **d** TRF analysis of wild type and independent late passage *MCM10*[+/−] cell populations (PD > 75). The location of peak intensity (yellow ovals) is indicated. **e** Genotyping of wild type (middle) or mutant alleles carrying a *loxP* site 3′ of exon 14 (upper) or a *loxP* scar (lower). A faint non-specific band is noted (asterisk). **f** Genotyping in populations analyzed in Fig. 3d showing alleles with one 3′ *loxP* site (upper) or a *loxP* scar (lower), and reverted alleles that retained or lost the 3′ *loxP* site. A faint non-specific band is noted (asterisk). **g** TRF analysis (top) and genotyping (bottom) in wild type and *MCM10*[+/−] cells. PDs and location of peak intensity (yellow ovals) are indicated. **h** Western blot for MCM10 with quantification normalized to tubulin, relative to the first lane is indicated. PDs for each cell line are noted. **i** Average proliferation rate in wild type, *MCM10*[+/−] and reverted cells normalized to early passage wild type cells, *n* = 6. Individual data points are indicated (gray circles). **j** Average cell cycle distribution of wild type, *MCM10*[+/−] and reverted cell lines, *n* = 4. Individual data points are indicated (gray circles). Percentage of each population in G1- (green), S- (purple), and G2/M-phase (gray) is indicated. Error bars in **h** and **i** indicate SD and significance was calculated using an unpaired, two-tailed student's *t-test* with \*<0.05; \*\*<0.01, \*\*\*<0.001. Source data for panels **a**, **b**, **c**, **d**, **e**, **f**, **g**, **h**, **i**, and **j**, including relevant exact p-values, are provided in the Source Data file.

origin licensing and DNA synthesis were significantly higher in wild-type HCT116 than in wild-type RPE-1 cells (Fig. 4j, k). Presumably, these data reflect differences between cell types, including changes that HCT116 cells underwent during oncogenic transformation, and suggest that highly proliferative transformed cells may be inherently more sensitive to MCM10 deficiency.

We hypothesized that the slowed proliferation and increased cell death in *MCM10* mutant RPE-1 cells were partially due to defects in telomere maintenance, as global changes in DNA synthesis were not observed (Fig. 4h). First, we measured β-gal activity to determine whether *MCM10*[+/−] cells activated senescence pathways. However, we did not detect any increase in *MCM10* mutants (Fig. 5a). When we monitored telomere length in wild type and *MCM10*[+/−] cells at regular intervals, we found that telomeres in *MCM10*[+/−] cells were shorter than wild type (Fig. 5b). Surprisingly, we found that telomeres in both cell lines elongated over time (Fig. 5b). This phenotype was telomerase-dependent, as passaging cells with the telomerase inhibitor BIBR1532[35] prevented elongation (Fig. 5c). These data suggested that telomerase activity in RPE-1 cells was robust enough to catalyze telomere extension, regardless of *MCM10* status. Moreover, although telomeres in wild-type RPE-1 cells were eroded slightly in the presence of inhibitor (~7 bp per PD), the identical treatment caused the rate of erosion in *MCM10*[+/−] mutants to double (~16 bp per PD; Fig. 5c). Thus, MCM10 deficiency also affected telomere length regulation in RPE-1 cells. To confirm this, we measured average telomere length in RPE-1 cells that carried the homozygous missense (c.1276 C > T, p.R426C) or heterozygous nonsense (c.1744C > T, p.R582X) NKD-associated variants, respectively. Remarkably, each *MCM10* mutant cell line carried telomeres that were significantly shorter than wild-type RPE-1 at similar passage (Fig. 5d), and the extent of telomere erosion in mutant cells corresponded with the severity of each *MCM10* variant. Taken together, our data is consistent with the notion that MCM10 deficiency limited telomerase-dependent telomere elongation in RPE-1 cells.

**Elevated telomerase activity rescues telomere length but not the inherent replication defect in heterozygous *MCM10* mutant cells**. To further evaluate the relationship between telomere length, MCM10 deficiency, and telomerase activity, we cultured wild-type and *MCM10* mutant HCT116 cells in the presence of telomerase inhibitor. TRF analyses showed that telomere erosion was exacerbated by telomerase inhibition (Fig. 6a), suggesting that MCM10 deficiency limited, but did not abolish telomerase-dependent elongation. The rate of attrition was similar for wild-type and MCM10 deficient cells (~20 to 25 bp per

PD), unlike our observations in RPE-1 cells. Thus, telomerase inhibition was dominant to the effect of MCM10 deficiency on telomere length regulation, as telomere attrition in *MCM10* mutants was not exacerbated when telomerase was completely inhibited. Interestingly, the population of *MCM10*[+/−] clone #8 was nearly 100% genetically reverted to *MCM10*[+/+] at PD 70, but telomere length had not recovered (Fig. 6a). We continued to propagate this population with and without telomerase inhibitor. Without inhibitor, telomeres efficiently lengthened over time (Fig. 6b). However, with inhibitor present telomeres remained short (Fig. 6b), confirming that telomerase activity was essential for telomere length recovery in reverted *MCM10* cells.

To test whether stable telomerase overexpression could rescue telomere erosion and alleviate HCT116 *MCM10*[+/−] mutant phenotypes, we generated stable cell lines that overexpressed *hTERT* and *hTR* – so-called 'super-telomerase' (ST)[36]. Most ST cell lines showed significantly longer telomeres (>12 kb) than observed in normal HCT116 cells (~4 to 6 kb; Fig. 6c). ST cell lines with significantly extended telomeres and similar telomerase activities (Fig. 6d) were utilized for further experiments. ST expression in *MCM10*[+/−] cell lines did not increase MCM10 expression levels (Fig. 6e), nor rescue the diminished proliferation rate (Fig. 6f). Interestingly, clonogenic survival decreased for wild-type ST cells, suggesting that long telomeres have a negative impact on proliferation of single-cell clones (Fig. 6g, Supplementary Fig. 5a). However, no significant difference was observed when comparing non-ST with ST MCM10-deficient cell lines (Fig. 6g, Supplementary Fig. 5a). Thus, although long telomeres may further reduce survival in both wild type and *MCM10*[+/−] ST cell lines, telomerase overexpression appears to compensate in *MCM10* mutants by rescuing cell death due to critically short telomeres. This balancing effect results in no discernable change in net survival. Overall, these data demonstrated that ST expression rescued telomere length without alleviating the underlying DNA replication defect.

The ST cells allowed us to perform telomeric replication assays similar to those used in models with long telomeres[26]. We utilized DNA combing to specifically analyze telomere and sub-telomere replication (Fig. 7a). In comparison to wild-type ST cells, *MCM10*[+/−] ST mutants showed an increase in unreplicated telomeres (Fig. 7b). Moreover, telomeres in *MCM10*[+/−] ST mutants were more often partially replicated, which is consistent with increased fork stalling (Fig. 7c). The increase in partially replicated telomeres in *MCM10* mutants was not attributable to differences in average telomere length, as measurements of completely and partially replicated telomeres found these to be in an identical size range (Supplementary Fig. 5b). Next, we used 2-dimensional (2D) gel analyses to detect DNA intermediates associated with telomeric replication stress. A low intensity t-circle arc was observed in all ST cell lines

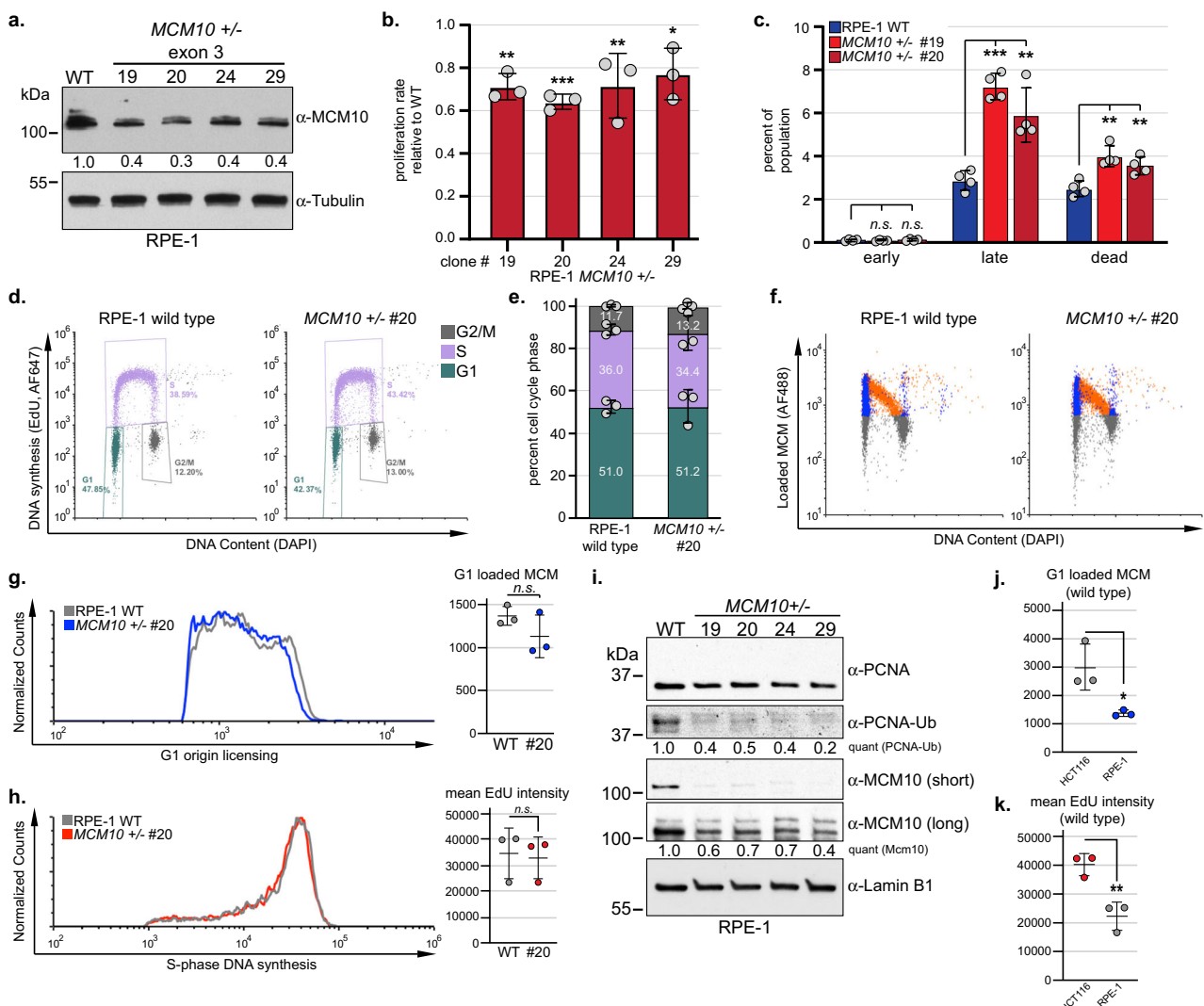

**Fig. 4 *MCM10* heterozygous RPE-1 cell lines have reduced MCM10 expression, impaired proliferation, and fewer active replication forks. a** Western blot for MCM10 with quantification normalized to tubulin relative to wild type is indicated. **b** Average proliferation rate in *MCM10*⁺/⁻ cells normalized to wild type. For each cell line $n = 6$ replicates. Average values for biological replicates are indicated (gray circles). **c** Average percentage of early apoptotic, late apoptotic, or dead cells. Wild type (blue) and clonal *MCM10*⁺/⁻ cell lines (red) are shown. For each cell line $n = 4$. Individual data points are indicated (gray circles). **d** Cell cycle distribution of wild type and *MCM10*⁺/⁻ cells. Percentage of each population in G1- (green), S- (purple), and G2/M-phase (gray) is shown. **e** Average cell cycle distribution of wild type and *MCM10*⁺/⁻ cells from three biological replicates. Percentage of each population in G1- (green), S- (purple), and G2/M-phase (gray). Individual data points are indicated (gray circles). **f** Flow cytometry plots for wild type and *MCM10*⁺/⁻ cells. G1-phase/MCM positive cells (blue), S-phase/MCM positive cells (orange) and G1- or G2/M-phase/MCM negative cells (gray) are indicated. **g** Comparison of origin licensing (left) and average G1 loaded MCM2 in biological replicates ($n = 3$) of wild type (gray) and *MCM10*⁺/⁻ cells (blue). **h** Comparison of S-phase DNA synthesis (left) and average EdU intensity in biological replicates ($n = 3$) of wild type (gray) and *MCM10*⁺/⁻ cell lines (red). **i** Chromatin-associated PCNA, PCNA-Ub MCM10, and Lamin B1. Quantification of PCNA-Ub or MCM10 levels normalized to Lamin B1, relative to the first lane is indicated. **j** Average G1 loaded MCM2 in biological replicates ($n = 3$) of wild type HCT116 (gray) and RPE-1 (blue) cells. **k** Average EdU intensity in biological replicates ($n = 3$) of wild type HCT116 (red) and RPE-1 (gray) cells. Error bars in **b**, **c**, **e**, **g**, **h**, and **j**, **k** indicate SD and significance was calculated using an unpaired, two-tailed student's *t*-test with *<0.05; **<0.01, ***<0.001. Source data for panels **a**, **b**, **c**, **e**, **g**, **i**, **j**, and **k**, including relevant exact p-values, are provided in the Source Data file.

(Fig. 7d). Strikingly, however, *MCM10*⁺/⁻ ST cells significantly accumulated t-complex DNA, which is comprised of branched DNA structures containing internal single-stranded (ss) DNA gaps (Fig. 7d)[37]. Treatment of wild-type HCT116 ST cells with hydroxyurea generated t-complex DNA (Fig. 7e), suggesting that these structures are products of replication stress. To further evaluate the nature of t-complex DNA, samples were treated with S1-nuclease to degrade ssDNA. Wild type ST samples were unchanged after S1-nuclease digestion, whereas a significant reduction in t-complex signal occurred in S1-digested *MCM10*⁺/⁻

ST samples (Fig. 7f). These data argued that the accumulated t-complexes in *MCM10* mutants were enriched for ssDNA gaps. Furthermore, the severity of chronic MCM10 deficiency in ST cells also stimulated spontaneous reversion that rescued the accumulation of t-complex DNA and the proliferation defect (Supplementary Fig. 5c, d). The generation of t-complexes is poorly understood, but their initial characterization was consistent with regressed forks and/or recombination intermediates[37]. To delineate between these possibilities, we measured the frequency of telomere sister chromatid exchanges (t-SCEs) that are produced by homologous

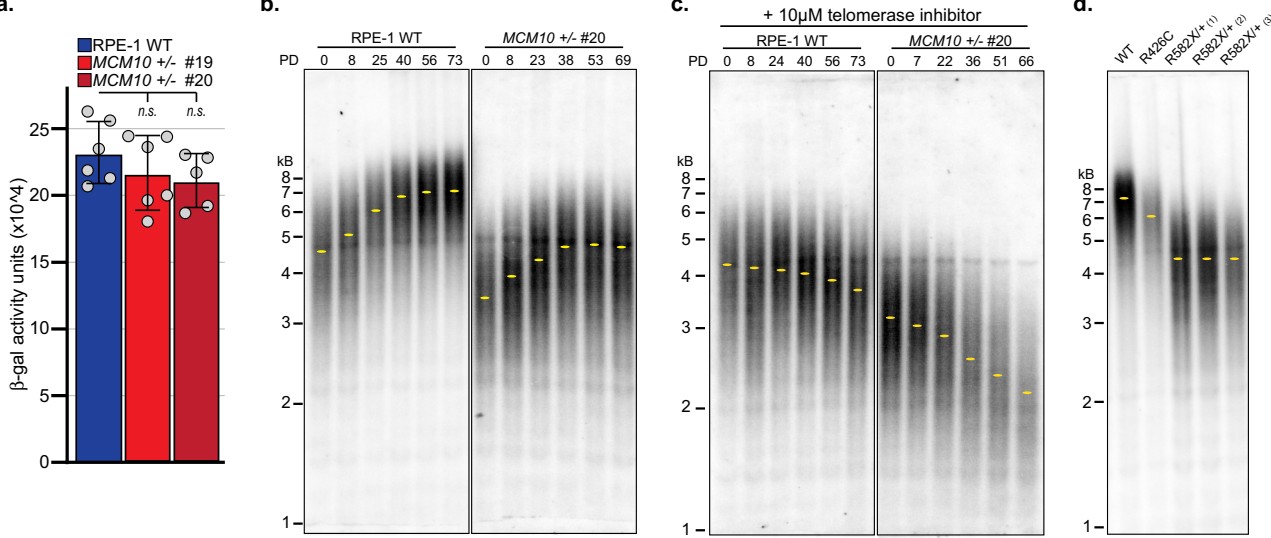

**Fig. 5 *MCM10* heterozygous RPE-1 cell lines have telomere maintenance defects. a** Average β-gal activity in wild type (blue) and clonal *MCM10^+/-* cell lines (red). For each cell line *n* = 6 replicates. Individual data points are indicated (gray circles). Error bars indicate SD and significance was calculated using an unpaired, two-tailed student's *t*-test with *<0.05; **<0.01, ***<0.001. **b** TRF analysis in wild type and *MCM10^+/-* clone #20 cell lines. PDs and location of peak intensity (yellow dots) are indicated. **c** TRF analysis in wild type and *MCM10^+/-* clone #20 cell lines in the presence of telomerase inhibitor. PDs and location of peak intensity (yellow dots) are indicated. **d** TRF analysis in wild type and *MCM10* mutant cell lines carrying NK-associated patient variants. Yellow dots indicate the location of peak intensity. Source data for panels **a–d**, including relevant exact p-values, are provided in the Source Data file.

recombination. *MCM10^+/-* ST cells showed a similar frequency of t-SCEs as wild-type ST cells (Fig. 7g). These data suggested that t-complexes in MCM10-deficient cells were not generated by homologous recombination but are the product of stalled telomeric replication forks.

**Loss of Mus81 exacerbates viability and telomere maintenance defects in Mcm10-deficient cells.** One mechanism to resolve fork stalling is the recruitment of structure-specific endonucleases (SSEs) to cleave replication intermediates and stimulate restart. A major player in this pathway is the mutagen-sensitive 81 (MUS81) endonuclease, which functions in complex with essential meiotic SSE1 or 2 (EME1 or EME2), as well as a larger DNA-repair tri-nuclease complex[38]. We hypothesized that *MCM10* mutants rely on SSE-dependent fork cleavage to overcome fork stalling. To test this, we generated *MCM10^+/-:MUS81^-/-* double mutants. Wild type and *MCM10^+/-* cells expressed equivalent MUS81 protein levels, whereas MUS81 was not detectable in *MUS81^-/-* mutants (Fig. 8a). Importantly, MCM10 expression was not altered following the knockout of *MUS81* (Fig. 8a). *MUS81^-/-* cells showed a growth defect, although not as severe as seen in *MCM10^+/-* mutants (Fig. 8b). Furthermore, this defect was significantly worse in the double mutants, which proliferated ~2-fold slower than *MCM10^+/-* single mutants (Fig. 8b). Comparison of clonogenic survival yielded similar results, whereby *MUS81^-/-* cells showed a colony formation defect that was less severe than *MCM10^+/-* mutants and *MCM10^+/-: MUS81^-/-* cells showed a stronger phenotype than either single mutant (Fig. 8c). Increased apoptosis or cell death was not detected in *MUS81* single mutants (Fig. 8d). However, we measured a significant increase in double-mutant cell death (Fig. 8d). Next, we utilized TRF analysis to examine alterations in telomere maintenance. *MUS81* knockout caused telomere erosion, and the loss of MUS81 in *MCM10^+/-* cells caused more significant erosion than in either single mutant. Furthermore, we found elevated β-gal signal in *MUS81* knockouts, with the highest levels detected in the double mutants (Fig. 8f). Taken together, our data clearly demonstrate a requirement for MUS81 in promoting cell proliferation and

viability in MCM10-deficient cells, likely through stimulating replication restart in hard-to-replicate regions, including telomeres.

## Discussion

**MCM10 deficiency causes genome instability and impairs telomere maintenance.** We have demonstrated that *MCM10* is haploinsufficient in HCT116 and hTERT-immortalized RPE-1 cell lines. Although a single loss-of-function allele does not cause human disease, the cellular phenotypes we describe here elucidate the molecular defects underlying the pathologies in patients carrying bi-allelic *MCM10* mutations. Both cell types showed a significant reduction in proliferation rate (Figs. 1f and 4b) likely caused by a decrease of active replication forks (Figs. 2j and 4i). It is noteworthy that MCM10 deficiency caused increased cell death regardless of cell type (Figs. 1i and 4c) and that this chronic replication stress was severe enough to stimulate spontaneous reversion of the exon 14 mutation and rescue of *MCM10* mutant phenotypes in HCT116 cells (Fig. 3f–j, Supplementary Fig. 5c, d). Our analyses of HCT116 mutants also demonstrated that *CDC45* and *MCM4* are haploinsufficient but suggested that a role in telomere maintenance is unique to *MCM10* (Supplementary Fig. 4).

Besides its well-defined role in origin activation, there is growing consensus that MCM10 promotes replisome stability[7,8,10], although the mechanism has remained unclear. A recent study demonstrated that yeast Mcm10 is important for bypassing lagging strand blocks[9], suggesting that Mcm10 is critical when the replisome encounters barriers to CMG translocation. Recently, yeast Mcm10 was shown to prevent translocase-mediated fork regression[39]. If human MCM10 functions similarly, it might not only prevent stalling but also inhibit excessive fork regression in order to promote restart. In support of this model, we argue that t-complex DNA in *MCM10* mutants was produced by regression of stalled telomeric replication forks. Without sufficient MCM10, cells must rely on alternative pathways to restart DNA replication. Our data implicate SSEs, including the MUS81 protein, in processing these

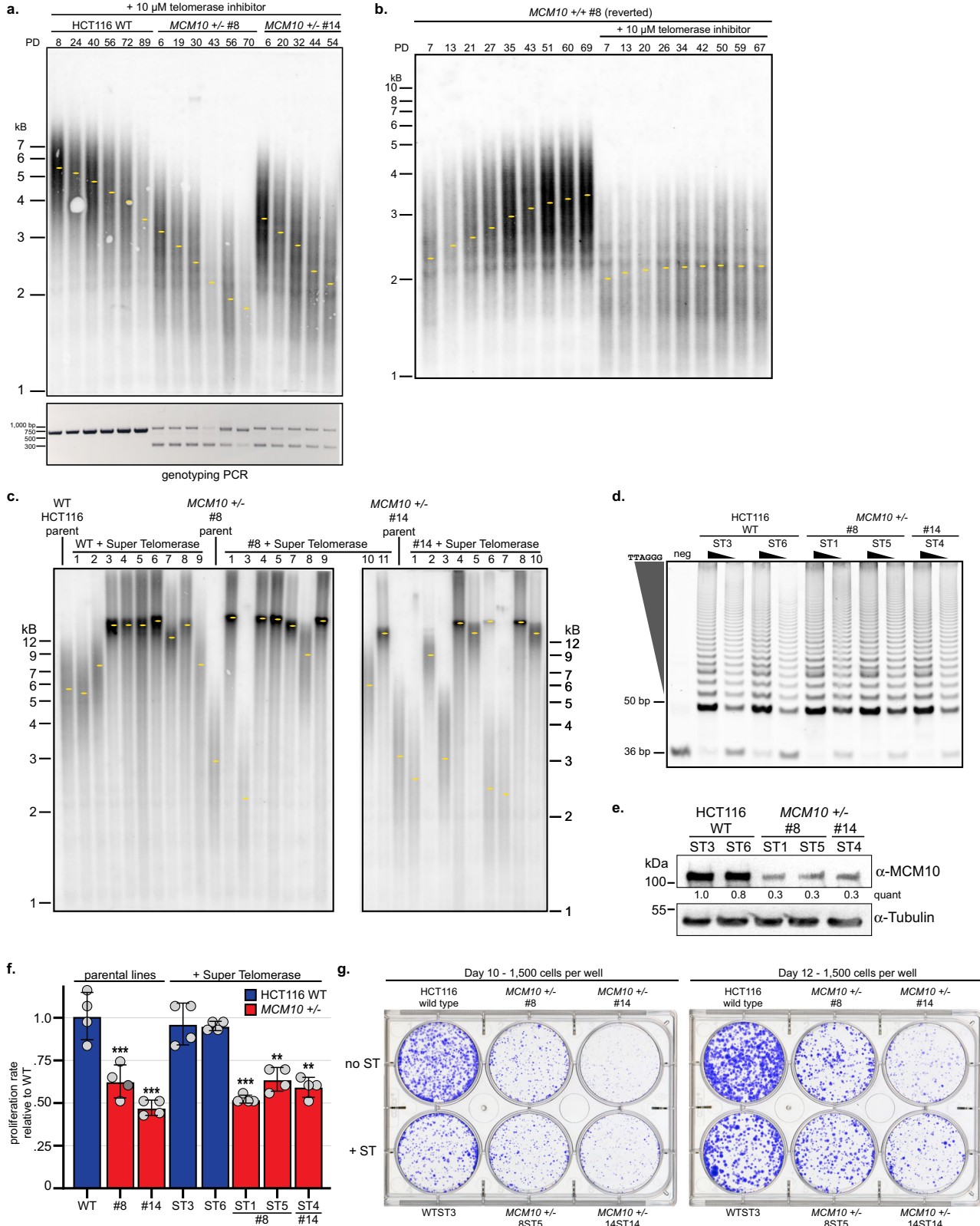

forks to rescue DNA synthesis and promote $MCM10^{+/-}$ cell viability (Fig. 8).

Defective telomere maintenance in *MCM10* mutants was not due to a decrease in intrinsic telomerase activity (Figs. 3c and 5d), but is a feature of genome instability caused by the underlying replication defects. In HCT116 *MCM10* mutants, we consistently observed telomeres that were shorter than in wild-type cells and

which did not recover unless *MCM10* expression was restored. Without restoration of *MCM10* expression, telomere length defects could be overcome with sufficiently high telomerase expression (Fig. 6c), although the underlying replication defects persisted. We propose that fork stalling prevented the replisome from reaching chromosome ends, thereby causing telomere shortening through several potential mechanisms. First, telomere

**Fig. 6 Overexpression of telomerase lengthens telomeres but does not rescue growth and viability defects. a** TRF analysis in HCT116 wild type and *MCM10*$^{+/-}$ cells (top) in the presence of 10 μM telomerase inhibitor BIBR1532. PDs and location of peak intensity (yellow dots) are indicated. Genotyping PCR for each time point is shown (bottom). **b** TRF analysis in *MCM10*$^{+/-}$ clone #8 reverted cells in the presence or absence of telomerase inhibitor. PDs and location of peak intensity (yellow dots) are indicated. **c** TRF analysis in HCT116 wild type, *MCM10*$^{+/-}$ cell lines, and ST derivatives of each parental cell line. **d** TRAP assay of HCT116 wild type (left) and *MCM10*$^{+/-}$ ST cell lines (right). The internal PCR control at 36 bp and telomerase products beginning at 50 bp are noted. For each cell line, two concentrations of cell extract were used representing a 10-fold dilution. **e** Western blot analysis of whole-cell extracts from HCT116 ST wild type and *MCM10*$^{+/-}$ cell lines for MCM10 with tubulin as a loading control. Quantification of MCM10 levels normalized to loading control, relative to the first lane wild-type ST3 sample is indicated. **f** Average proliferation rate in HCT116 wild type, *MCM10*$^{+/-}$ and ST cell lines normalized to wild type cells. For each cell line *n* = 4; error bars indicate SD and significance was calculated using an unpaired, two-tailed student's *t*-test with *<0.05; **<0.01, ***<0.001. Individual data points are indicated (gray circles). **g** Comparison of clonogenic survival in HCT116 wild type, *MCM10*$^{+/-}$ and ST cell lines after 10 or 12 days in culture. The number of cells plated per well is indicated. Source data for panels **a**, **b**, **c**, **d**, **e**, and **f**, including relevant exact p-values, are provided in the Source Data file.

replication defects might delay the generation of chromosome ends suitable for telomerase extension. Because telomere replication and telomerase-dependent synthesis primarily occur later in S-phase[40–42], the replication defect could delay telomere maturation and shorten the cell cycle window when telomerase can act. Second, because reversed replication forks are an aberrant substrate for telomerase[43], increased fork stalling and regression could impair normal telomerase recruitment or activity. Third, terminal fork arrest in the proximal telomere requiring nucleolytic cleavage to form a functional chromosome end could result in the loss of a significant portion of the distal telomere.

**Development of distinct cell lineages is impaired by *MCM10*-associated telomeropathy.** Prior investigations of *MCM10*-associated disease have primarily focused on cancer development. Many studies have described *MCM10* overexpression in a variety of cancer types, wherein the extent of upregulation corresponded with tumor progression and negative clinical outcomes[7]. Presumably, increased MCM10 expression served to prevent DNA damage and maintain genome stability in cells driven to proliferate. In addition, our data argue that telomere length maintenance in transformed cells, a widely accepted hallmark of cancer, relies on MCM10[44]. This increased demand in cancer cells also suggests that compounds inhibiting MCM10 might be useful to preferentially sensitize cancer cells to treatment with common chemotherapeutic drugs or telomerase inhibitors[45].

We have reported one *MCM10*-associated case of NKD[12]. These variants produced phenotypes similar to the genomic instability caused by NKD-associated variants in replication factors *MCM4* and *GINS1*[46–48]. It is currently unknown whether these *MCM4* or *GINS1* patient variants affect telomere maintenance. However, another NKD patient was identified carrying a variant in regulator of telomere length 1 (*RTEL1*)[48,49], an essential DNA helicase required for the replication of mammalian telomeres[25]. Interestingly, MCM10 was identified as an RTEL1-interacting protein in mass spectrometry analyses of murine cells[50], supporting the hypothesis that telomeropathy might underlie NKD. Limited evidence suggests that NK cells have shorter telomeres than T- and B-cells, although they arise from a common progenitor, and that telomerase activity decreases as NK cells differentiate[51–53]. These features may explain why the NK cell lineage is particularly sensitive to defects in telomere replication.

Here, we report *MCM10*-associated RCM with lymphoreticular hypoplasia caused by bi-allelic *MCM10* variants that resulted in a remarkably similar disease phenotype in three affected patients. To our knowledge, this represents the first connection between alterations in core replisome function and inherited cardiomyopathy. However, cardiomyopathy has been linked with defective telomere maintenance. For example, recent studies have argued that telomere shortening is a hallmark of inherited

cardiomyopathies and that long telomeres may be cardioprotective[54,55]. Additionally, cardiomyopathy has been recognized as a feature of some cases of the telomeropathy dyskeratosis congenita (DC)[56–58], although these cases are rare. Intriguingly, variants in *RTEL1* – discussed above in the context of NKD – have also been strongly linked with telomeropathies including DC[59,60], demonstrating that variants in the same gene can give rise to clinically distinct pathologies. We propose that this is also the case for *MCM10* and that telomere erosion due to MCM10 deficiency was the cause of cardiomyopathy in these patients.

The absence of pathology in family members carrying a single *MCM10* variant demonstrates that mono-allelic null mutations are not haploinsufficient in human beings. Because each heterozygous combination included one null allele, the hypomorphic alleles in each pair seem necessary to further reduce MCM10 function and elicit disease. Whereas the NKD-associated missense allele caused a single amino acid substitution, the RCM-associated splice donor variant appears to be more deleterious, causing exon skipping, reduced expression of wild type *MCM10* mRNA and potential accumulation of a protein variant that carries an internal deletion affecting the major MCM10 DNA-binding domain[7] (Supplementary Fig. 1b-d). It is striking that both compound heterozygous *MCM10* variants caused immune system abnormalities, albeit to different extents. Furthermore, the more severe combination of alleles (c.236delG;764 + 5 G > A) also impaired cardiac development and is believed to have been lethal in utero. In contrast, the NKD-associated alleles (c.1276 C > T;1744C > T) resulted in a live-born infant with no overt cardiac phenotype. Based on these observations, we propose that the threshold for MCM10 function necessary for normal tissue development is cell lineage specific and that differences in the severity of *MCM10* variants explain why patients presented with distinct but overlapping pathologies (Fig. 8g). Within this model of *MCM10*-associated telomeropathy, one key factor that determines the MCM10 threshold that still allows for normal differentiation is the rate of telomere shortening inherent to the normal development of each specific cell lineage. Therefore, different lineages have their own intrinsic thresholds, explaining the cell type-specific clinical presentations. Future analysis of different human cell lineages and/or separation-of-function *MCM10* mutations would be required to demonstrate that MCM10-dependent telomere maintenance defects are causative in these human diseases.

## Methods
**Patient participation.** Details of the consent process and qualitative analysis of decision-making in the genomic medicine multi-disciplinary team (GM-MDT) have been described including how, dependent on consent, patients had the option to receive "secondary findings"[61]. Patients participated under the Molecular Genetic Analysis and Clinical studies of Individuals and Families at Risk of Genetic

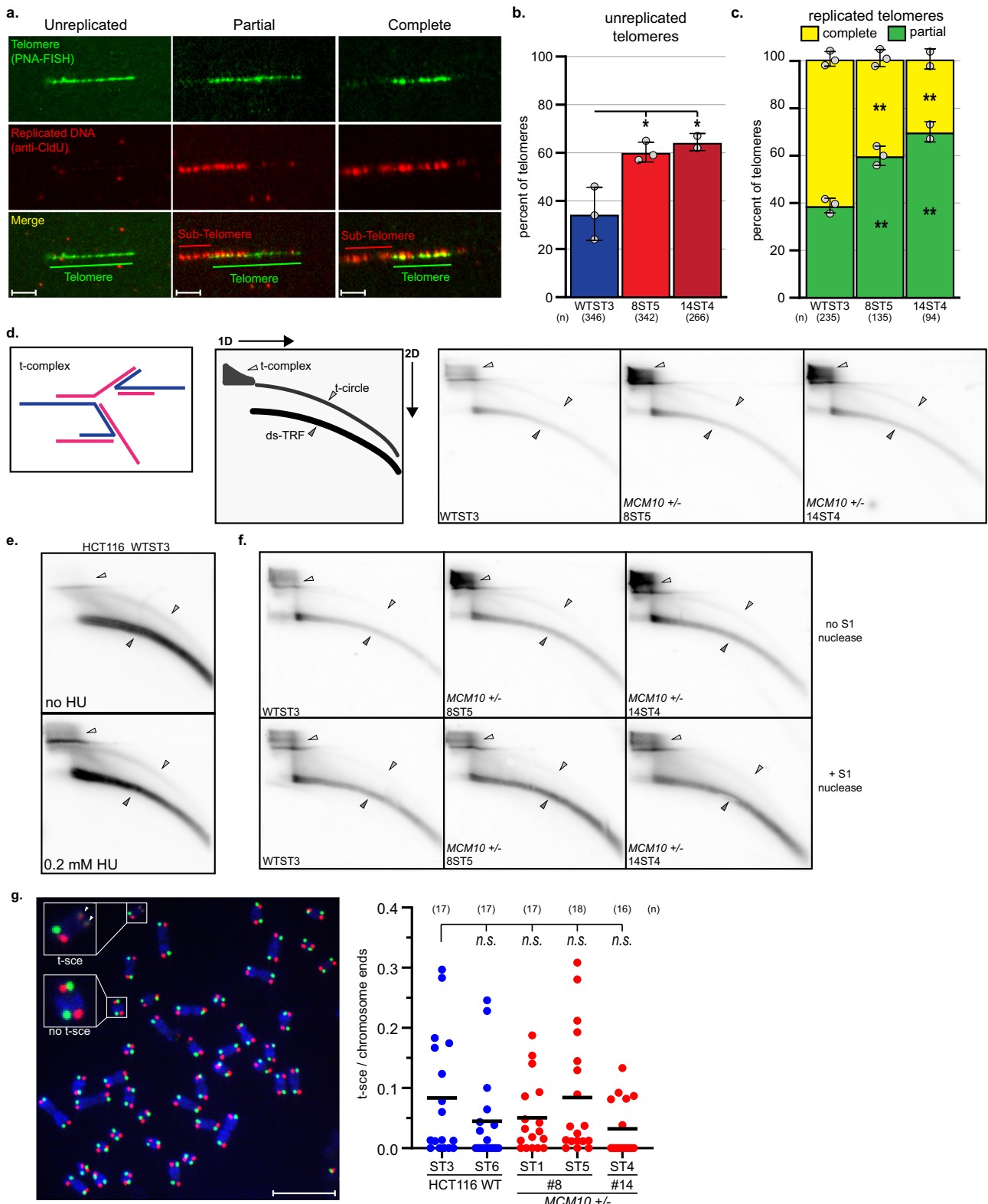

Disease (MGAC) protocol approved by West Midlands Research Committee, reference number 13/WM/0466.

**Clinical samples**. Following written informed consent for genetic testing from the patient and/or their parent, blood or post mortem splenic tissue was obtained for genomic DNA extraction. Clinical samples were processed and sequencing results were validated in the Oxford Regional Clinical Molecular Genetic Laboratory.

Exome sequencing was performed at the Wellcome Centre for Human Genetics, Oxford. Whole genome sequencing was performed in the BRC/NHS Molecular Diagnostics Laboratory of the Oxford University Hospitals Trust.

**Exome and whole genome sequencing, and bioinformatic analyses**. Exome sequencing[62] was performed using DNA libraries prepared from 3 μg patient DNA extracted from whole blood (parents) or spleen (proband). Exome capture was

**Fig. 7 Telomeric replication stress is increased in ST *MCM10*$^{+/-}$ HCT116 cells. a** Telomere combing images with telomeric DNA (green), nascent DNA (red), and merged images with examples of unreplicated telomeres (left), partially replicated (middle), and completely replicated telomeres (right). Telomeric and sub-telomeric regions are indicated. Scale bars are 3 µm. **b** Average percentage of unreplicated telomeres in wild type (blue) and *MCM10*$^{+/-}$ cell lines (red) ST cell lines, $n =$ total number of telomeres quantified. Average values for each replicate are indicated (gray circles). **c** Average percentage of completely replicated (yellow) versus partially/stalled telomeres (green) in ST cell lines, $n =$ total number of replicated telomeres quantified. Average values for each replicate are indicated (gray circles). Error bars in **b** and **c** indicate SD and significance was calculated using an unpaired, two-tailed student's *t*-test with *<0.05; **<0.01; ***<0.001. **d** (Left) Cartoon of t-complex DNA as formerly depicted[37]. (Middle) Diagram of double-stranded telomere restriction fragment (ds-TRF), telomere circle (t-circle) and telomere complex (t-complex) DNA species from 2D gel electrophoresis. (Right) Comparison of 2D gels from ST cell lines. **e** Comparison of DNA species from 2D gel electrophoresis in HCT116 wild-type ST cells with and without 4-day HU treatment. **f** Comparison of DNA species from 2D gels in HCT116 wild type and *MCM10*$^{+/-}$ ST cell lines with and without S1 nuclease digestion. **g** (Left) Image of t-SCE staining in ST cell lines. Examples of a t-SCE event and chromosomes without t-SCE are highlighted. (Right) Percentage t-SCE per chromosome ends. Bars represent average percentage t-SCE per chromosome ends; where $n =$ the number of metaphases analyzed. Significance was calculated using an unpaired, two-tailed student's *t*-test but were not statistically significant (*n.s.*). Scale bar is 10 µm. Source data for panels **b**, **c**, **d**, **e**, **f**, and **g**, including relevant exact p-values, are provided in the Source Data file.

performed using SeqCap EZ Human Exome Library v2.0 (NimbleGen), according to the manufacturer's instructions, and sequenced using a 100 bp paired-end read protocol on the HiSeq2500 (Illumina). Exome sequence reads were mapped to the hs37d5 reference genome with Stampy version 1.0.20[63]. Variants were called with Platypus version 0.5.2[64]. Variants were annotated and analyzed using Variant-Studio version 2.2 (Illumina) and Ingenuity VA (Qiagen). Initial data analysis focused on variants associated with an in silico panel of 53 cardiomyopathy-associated genes, but was later extend to all genes.

Whole-genome sequencing was performed in the HICF2 project[65,66] using DNA libraries prepared from 3 µg patient DNA extracted from spleen. Sequencing was performed using a 100 bp paired-end read protocol on the HiSeq2500 (Illumina). Sequence reads were mapped to the hs37d5 reference genome with Stampy version 1.0.20[63]. Variants were called with Platypus version 0.5.2[64]. Variants were annotated and analyzed using VariantStudio version 2.2 (Illumina) and Ingenuity VA (Qiagen). Variant analysis focused on autosomal recessive and X-linked modes of inheritance. For copy-number detection Log$_2$R values were generated from bam files and were analyzed together with B-allele frequency outputs. Events were flagged and visualized using Nexus Discovery Edition (BioDiscovery, Inc., El Segundo, CA). The *MCM10* variants were independently validated by Sanger sequencing using BigDye Terminator kit 3.1 (Applied Biosystems) combined with purification using the Agencourt CleanSEQ system. Capillary electrophoresis was performed using an ABI Prism 3730 Genetic Analyser (Applied Biosystems). Effects of the c.764 + 5 G > A variant were assessed using RT-PCR with standard procedures and PCR primers positioned within *MCM10* exons 3 and 8. Analysis of splice site strength was performed using MaxEntScan[14].

**Protein expression and purification.** The coding sequence for human MCM10 internal domain (ID) spanning Ser236 to Gly435, with a preceding HRV 3 C protease recognition sequence (LEVLFQGP), was inserted into the NdeI/BamHI sites of the pET28a bacterial expression vector. To mimic the internal deletion of exon 6 and use of the exon 7 cryptic splice acceptor, a construct lacking the N-terminal 27 amino acid residues (Ser236 to Thr262) was generated and referred to as ΔN27. Recombinant wild-type and mutant (ΔN27) hMCM10-ID were expressed in *E. coli* strain BL21(DE3) and purified[45]. The total bacterial lysate was centrifuged at 48,384 × g at 4 °C for 1 h (Beckman JA-25.50 rotor), and the supernatant was applied to a 5 mL HisPur Ni-NTA resin (Thermo Fisher Scientific) equilibrated with 20 mM Tris-HCl, pH 7.4, 500 mM NaCl, 5 mM β-mercaptoethanol, 5 mM imidazole. Bound His-tagged proteins were subsequently eluted with a linear imidazole concentration gradient up to 300 mM. The eluted proteins were then treated overnight with recombinant HRV 3 C protease to remove the N-terminal His-tag and further purified via size exclusion chromatography on a HiLoad 26/600 Superdex 75 column (GE Healthcare) operating with 20 mM Tris-HCl, pH 7.4, 500 mM NaCl, 5 mM β-mercaptoethanol. Monomeric protein peaks were pooled and concentrated using an ultrafiltration device (Amicon) to 24.6 and 15.6 mg ml$^{-1}$ for wild type and ΔN27, respectively, flash cooled by liquid nitrogen in small aliquots and stored at −80 °C. Protein concentrations were determined based on UV absorbance measured on a Nanodrop 8000 spectrophotometer and theoretical extinction coefficient values calculated from the protein sequences. The purified proteins after HRV 3 C protease cleavage have two extra amino acids (Gly-Pro) on the N-terminus.

**Analytical size-exclusion chromatography.** Wild type or ΔN27 hMCM10-ID (200 µL at 5 mg ml$^{-1}$) was injected into a Superdex 75 Increase 10/300 GL column (GE Healthcare) operating with the elution buffer containing 20 mM Tris-HCl, pH 7.4, 500 mM NaCl, 5 mM β-mercaptoethanol, and a flow rate of 0.4 mL min$^{-1}$. The proteins were detected by UV absorption at 280 nm. To calibrate the column, we injected molecular weight standards (Bio-Rad) including thyroglobulin, γ-globulin, ovalbumin, myoglobin, and vitamin B12 under the same elution condition. All experiments were performed at 4 °C.

**Circular dichroism spectroscopy.** Circular dichroism (CD) spectra were collected on a Jasco J-815 CD spectropolarimeter with a temperature controller, using quartz cuvettes with a pathlength of 0.1 cm (Starna Ltd.). The samples contained purified protein at 0.1 mg mL$^{-1}$ in 50 mM sodium phosphate, pH 7.4 and 0.5 mM TCEP. The blank (baseline) sample contained the same buffer without protein. Data acquisition was performed for the wavelength range of 190 to 260 nm with 1 nm steps, at the temperatures of 25, 53, 60, 65, 70, 75, and 80 °C (only the data for 25 and 60 °C are shown). Fresh protein sample was prepared for the measurement at each temperature and the spectra were scanned five times. The data are presented as molar ellipticity ([q]/deg cm$^2$ dmol$^{-1}$ ×10$^5$) plotted versus wavelength. To monitor thermal denaturation, CD spectra for the wavelength range from 205 to 220 nm, in 1 nm steps, were collected over the temperature range from 40 to 90 °C in 1 °C increments. The spectra were scanned five times at each temperature following a 10-s equilibration period. Single wavelength melt curves were obtained by plotting the measured CD signal at 205 nm against temperature. The curves were normalized by setting the minimum (greatest magnitude) ellipticity at 40 °C as 0 and maximum ellipticity at 90 °C as 1. The temperature at which the normalized CD signal reaches 0.5 was defined as the melting temperature (T$_m$).

**Cell lines.** HCT116 cells were grown in McCoy's 5 A medium (Corning 10-050-CV) supplemented with 10% FBS (Sigma F4135), 1% Pen Strep (Gibco 15140), and 1% L-glutamine (Gibco 205030). hTERT RPE-1 cells were grown in DMEM/F12 medium (Gibco 11320) supplemented with 10% FBS and 1% Pen Strep. Cells were cultured at 37 °C and 5% CO$_2$.

**Cell line generation.** HCT116 *MCM10*$^{+/-}$ (exon 14) cell lines were generated using rAAV-mediated gene targeting[67]. The conditional vector pAAV-MCM10-cond was constructed using Golden Gate cloning[67]. The MCM10-cond rAAV was generated by co-transfection of pAAV-MCM10-cond, pAAV-RC[67], and pHelper[67] into HEK293 cells using Lipofectamine LTX (Invitrogen 15338030) following standard protocols[67]. The first round of targeting replaced *MCM10* exon 14 with a wild-type allele and a downstream neomycin (G418) selection cassette both flanked by *loxP* sites ("floxed"). Targeted clones were selected using 0.5 mg/ml G418 (Geneticin G5005). Resistant clones were screened by PCR using primers within the neomycin cassette and outside the rAAV homology arms to confirm locus-specific targeting, and Cre recombinase transiently expressed from an adenoviral vector (AdCre; Vector Biolabs #1045) was then used to remove the neomycin selection cassette as described[67]. The second round of *MCM10* gene targeting used the same rAAV vector and replaced the wild-type allele with a floxed allele and a downstream floxed neomycin selection cassette. G418-resistant clones were screened by PCR to confirm locus-specific targeting. AdCre recombinase was then used to remove the neomycin selection cassette and resulted in the generation of heterozygous *MCM10* clones. The *MCM10* exon 14 genotype was subsequently screened and confirmed using primers flanking the exon. To understand why the upstream *loxP* site was not incorporated, we assessed the probability for the two loxP sites to be incorporated into the flanking regions of exon 14 based on Kan et al.[68]. We determined that the integration probabilities for the upstream and downstream *loxP* sites are approximately 0.45 and 0.90, respectively. We screened 103 neomycin-resistant colonies, 14 of which tested positive by PCR for locus-specific targeting. We randomly selected four clones for further genotyping. Of these, two of them had incorporated both *loxP* sites, and two only had the downstream *loxP* site, consistent with the predicted probabilities. The clones that were lacking the *loxP* site upstream of exon 14 but contained a floxed NEO gene easily excised the neomycin marker after transient introduction of Cre-recombinase.

HCT116 and RPE-1 *MCM10*$^{+/-}$ exon 3, HCT116 *CDC45*$^{+/-}$ exon 3, HCT116 *MCM4*$^{+/-}$ exon 2, and HCT116 *MUS81*$^{-/-}$ exon 2 cell lines were generated using CRISPR/Cas9 gene targeting. Guide RNAs (gRNAs) were cloned into a CRISPR/Cas9 plasmid hSpCas9(BB)−2A-GFP (PX458; Addgene #48138)[69]. Cells were transfected with CRISPR/Cas9 plasmid containing gRNA using the Neon

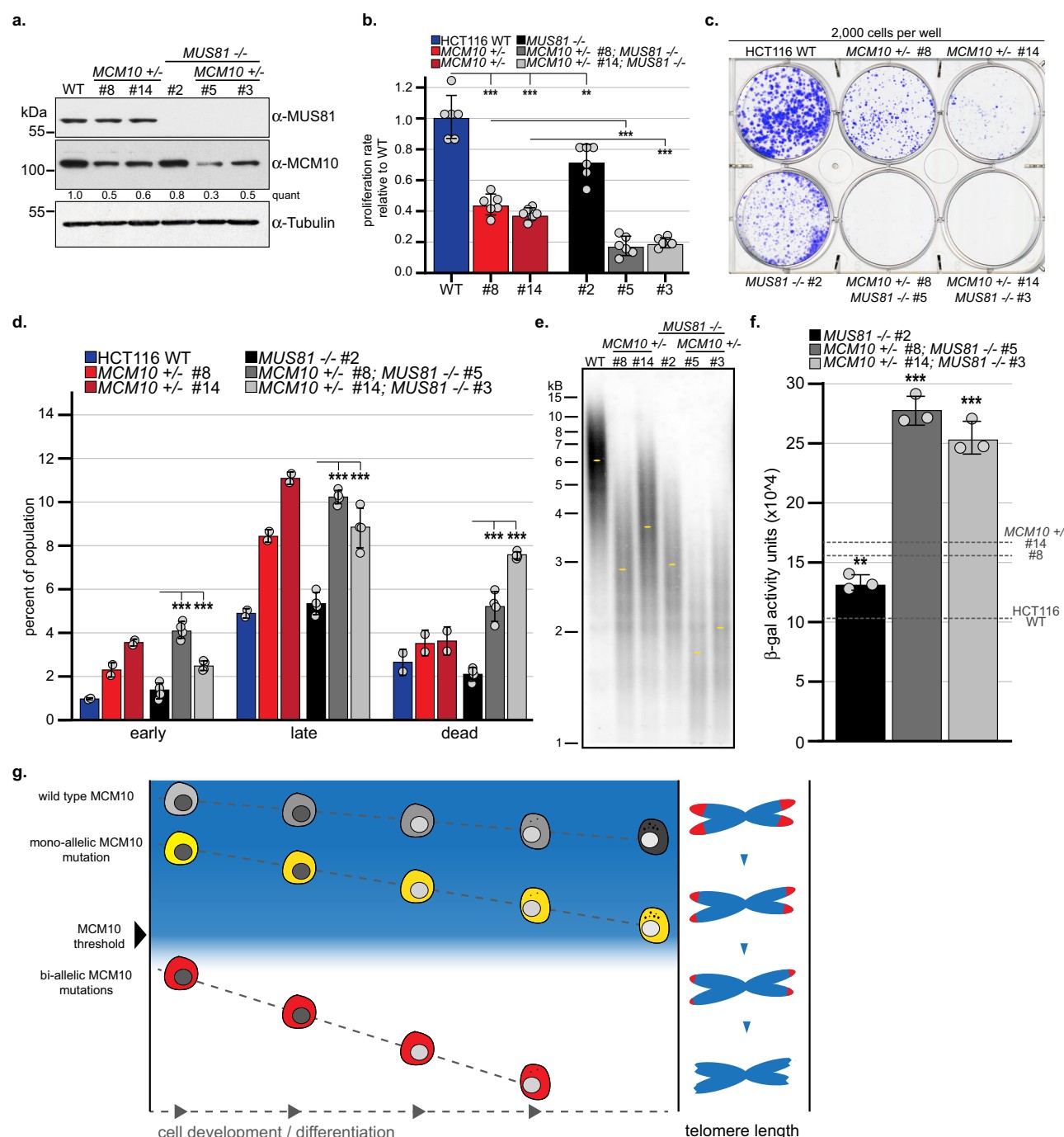

**Fig. 8 Loss of MUS81 increases the severity of proliferation, viability, and telomere length defects in _MCM10_$^{+/-}$ cell lines. a** Western blots for MUS81 and MCM10. Quantification normalized to tubulin relative to the wild type sample is indicated. **b** Average proliferation rate in HCT116 wild type, _MUS81_$^{-/-}$, _MCM10_$^{+/-}$ and double mutant cell lines normalized to HCT116 wild type, $n = 6$. Individual data points are indicated (gray circles). **c** Comparison of clonogenic survival of HCT116 wild type, _MUS81_$^{-/-}$, _MCM10_$^{+/-}$ and double mutant cell lines. **d** Average percentage of each population represented by early apoptotic, late apoptotic or dead cells in HCT116 wild-type, _MUS81_$^{-/-}$, _MCM10_$^{+/-}$ and double mutant cell lines, $n = 2$ replicates for HCT116 wild type and _MCM10_$^{+/-}$ single mutants; $n = 4$ for all _MUS81_$^{-/-}$ cell lines. Individual data points are indicated (gray circles). **e** TRF analysis of early passage HCT116 wild type, _MUS81_$^{-/-}$, _MCM10_$^{+/-}$ and double mutant cell lines. Location of peak intensity (yellow dots) is indicated. **f** β-gal activity expressed as arbitrary fluorescence units normalized to total protein for _MUS81_$^{-/-}$ (black) and _MUS81_$^{-/-}$, _MCM10_$^{+/-}$ mutant cell lines (gray). Average levels for HCT116 wild type and _MCM10_$^{+/-}$ cell lines from Supplementary Fig. 3d are indicated with dashed lines, $n = 3$. Individual data points are indicated (gray circles). Error bars in **b**, **d**, and **f** indicate SD and significance was calculated using an unpaired, two-tailed students $t$-test with *<0.05; **<0.01, ***<0.001. Source data for panels **a**, **b**, **d**, **e**, and **f**, including relevant exact p-values, are provided in the Source Data file. **g** Model of _MCM10_-associated telomeropathies. Different cell lineages have an inherent developmental threshold for MCM10 expression to achieve complete development and differentiation. As mono- or bi-allelic variants decrease the amount of functional MCM10, telomere erosion is accelerated. When MCM10 function is reduced below the required threshold, eroded telomeres cause replicative senescence and prevent complete development.

Transfection System (Invitrogen MPK5000) following standard protocols. Two days post-transfection GFP-positive cells were collected by flow cytometry. Subcloned cells were screened for correct targeting by PCR amplification and restriction enzyme digestion (*MCM10* exon 3, Hpy199III (NEB R0622); *CDC45* exon 3, PflMI (NEB R0509), XcmI (NEB R0533) or AlwNI (NEB R0514); *MCM4* exon 2, BglI (NEB R0143). To generate HCT116 *MUS81*[−/−] exon 2 mutant cell lines, each parental line was transfected with 1 µL of 100 µM sgRNA (Synthego Corporation) and 1 µg Cas9 mRNA (TriLink #L-7206) using the Neon Transfection System following standard protocols. Three days post-transfection cells were subcloned. Subclones were screened for correct targeting by PCR amplification and Illumina sequencing or PCR amplification, DNA sequencing, and Tracking of Indels by Decomposition (TIDE) analyses version 3.2.0[70]. A list of primers and oligos used for all cell line generation and validation is provided in Supplementary Table 2.

**Cell line generation using plasmid transfection**. To generate HCT116 cell lines expressing super-telomerase[36], pBABEpuroUThTERT+U3-hTR-500[71] (Addgene #27665) was linearized with restriction enzyme ScaI (NEB R3122) and transfected into wild type or mutant HCT116 cell lines following standard Lipofectamine 3000 protocols (Invitrogen L3000). Stable cell lines were generated using puromycin selection (1 µg/ml; Sigma P7255) followed by subcloning.

**Cell proliferation**. Cells were plated at 50,000 cells per well (RPE-1) or 100,000 to 125,000 cells per well (HCT116) in 6-well plates. Cell counts were performed 3-days after seeding using Trypan Blue (Invitrogen T10282) on Countess slides (Invitrogen C10283) using a Countess automated cell counter (Invitrogen C20181). For all experiments, *n* is represented by technical and/or biological replicates as described in each figure legend. Statistical analysis was performed using Microsoft Excel version 16.44.

**Protein extraction, chromatin fractionation, and western blotting**. For preparation of whole-cell extracts, cells were lysed in RIPA (50 mM Tris-HCl, pH 8.0, 150 mM NaCl, 10 mM NaF, 1% NP-40, 0.1% SDS, 0.4 mM EDTA, 0.5% sodium deoxycholate, 10% glycerol) buffer for 10 min and then centrifuged at 16,000 *g* for 10 min. Cleared lysates were collected, mixed with SDS loading buffer and boiled before fractionation by SDS-PAGE and analyses by western blot. Extracts were prepared[72] by lysis in Buffer A (10 mM HEPES pH 7.9, 10 mM KCl, 1.5 mM MgCl$_2$, 0.34 M sucrose, 10% glycerol, 0.1% Triton X-100 and protease inhibitors). Insoluble nuclear proteins were isolated by centrifugation and chromatin-bound proteins were subsequently released by sonication. The remaining insoluble factors were cleared by centrifugation before fractionation by SDS-PAGE and western blot analyses. Primary antibodies were incubated in 5% BLOT-QuickBlocker (G-Biosciences 786-011) as follows: rabbit anti-MCM10 (Bethyl A300-131A; 1:500), rabbit anti-MCM10 (Novus, H00055388-D01P, 1:500), mouse anti-CDC45 (Santa Cruz G12, SC55568; 1:500), mouse anti-MCM4 (Santa Cruz G7, SC28317; 1:500), mouse anti-MUS81 (Abcam ab14387; 1:500), mouse anti-PCNA (Abcam Ab29; 1:3,000), rabbit anti-Ubiquityl-PCNA (Lys164) (Cell Signaling Technology mAb13439, D5C7P; 1:1,000), rabbit anti-RPA32 (S4/8) (Bethyl A300-245A; 1:2000), mouse anti-GAPDH (GeneTex GTX627408; 1:5,000), mouse anti-α-Tubulin (Millipore T9026, clone DM1A; 1:10,000), rabbit anti-Lamin B1 (Proteintech #12987; 1:3,000). Secondary antibodies were incubated in 5% BLOT-QuickBlocker (G-Biosciences 786-011) at 1:10,000 dilutions, including goat anti-mouse HRP conjugate (Jackson Laboratories 115-035-003), goat anti-rabbit HRP conjugate (Jackson Laboratories 111-035-144), goat anti-mouse HRP conjugate (BioRad, 1706516), donkey anti-rabbit HRP conjugate (Amersham NA9340). Detection was performed using WesternBright Quantum detection kit (K-12042-D20). Quantification was performed using FIJI version 2.1.0/153.c and Microsoft Excel version 16.44. Image preparation was performed using Adobe Photoshop version 21.2.1.

**FACS analysis**. For flow cytometry analyses[18] of cell cycle, DNA synthesis and origin licensing, wild type and *MCM10*[+/−] HCT116 and hTERT RPE-1 cell lines were incubated with 10 µM EdU (Santa Cruz sc-284628) for 30 min before harvesting with trypsin. Soluble proteins were extracted in CSK (10 mM PIPES pH 7.0, 300 mM sucrose, 100 mM NaCl, 3 mM MgCl$_2$ hexahydrate) with 0.5% Triton X-100, then cells were fixed in PBS with 4% PFA (Electron Microscopy Services) for 15 min. Cells were labeled with 1 µM AF647-azide (Life Technologies A10277) in 100 mM ascorbic acid, 1 mM CuSO$_4$, and PBS to detect EdU for 30 min, at room temperature. Cells were washed, then incubated with MCM2 antibody 1:200 (BD Biosciences #610700) in 1% BSA in PBS with 0.5% NP-40 for 1 hr at 37 °C. Next, cells were washed and labeled with donkey anti-mouse AF488 secondary antibody 1:1,000 (Jackson Immunoresearch 715-545-150) for 1 hr at 37 °C. Lastly, cells were washed and incubated in DAPI (Life Technologies D1306) and 100 ng/mL RNase A (Sigma R6513) overnight at 4 °C. For all experiments, *n* is represented by biological replicates. Samples were gated and analyzed[18] on an Attune NxT (Beckman Coulter) or LSR II (BD Biosciences) flow cytometer and analyzed with FCS Express version 6 (De Novo Software) or FlowJo version 10.6.1 and Microsoft Excel version 16.44.

For flow cytometry analysis of apoptosis, cells were seeded in 6-well plates (HCT116 wild-type at 150,000 cells/well, HCT116 mutants at 200,000 to 400,000

cells/well, RPE-1 wild-type at 50,000 cells/well, and RPE-1 mutants at 75,000 cells/well) and allowed to proliferate for approximately 72 hr. Adherent and floating cells were collected, washed with 1x PBS twice, and stained using the APC Annexin V apoptosis detection kit (Biolegend 640932) according to the manufacturer's instructions. Samples were analyzed on a FACSCanto A V0730042 (BD Biosciences). Apoptotic cells were identified by annexin V staining while cell viability was determined by PI staining. For all experiments, *n* is represented by biological replicates. Data was analyzed using FlowJo version 10.6.1 and Microsoft Excel version 16.44. The gating strategy for this technique is included in Supplementary Fig 2d.

**DNA combing**. HCT116 cells were plated at 1×10$^6$ cells per 10 cm plate 48 hr prior to labeling. Cells were incubated with 25 µM IdU (Sigma C6891) for 30 min, rinsed with pre-warmed medium and then incubated with 200 µM CldU (Sigma I7125) for 30 min. Approximately 250,000 cells were embedded in 0.5% agarose plugs (NuSieve GTG Agarose, Lonza, 50080) and digested for 48 to 72 hr in plug digestion solution (10 mM Tris-HCl, pH 7.5, 1% Sarkosyl, 50 mM EDTA and 2 mg/ml Proteinase K). Plugs were subsequently melted in 50 mM MES pH 5.7 (Calbiochem #475893) and digested overnight with β-agarose (NEB M0392). DNA combing was performed using commercially available coverslips (Genomic Vision COV-001). Integrity of combed DNA for all samples was checked by staining with YOYO-1 (Invitrogen Y3601). Combed coverslips were baked at 60 °C for 2 to 4 hr, cooled to room temperature (RT) and stored at −20 °C. DNA was denatured in 0.5 M NaOH and 1 M NaCl for 8 min at RT. All antibody staining was performed in 2% BSA in PBS-Triton (0.1%). Primary antibodies include rabbit anti-ssDNA (IBL 18731; 1:5), mouse anti-BrdU/IdU (BD Biosciences 347580; 1:10) and rat anti-BrdU/CldU (Abcam ab6326; 1:20). Secondary antibodies include goat anti-mouse Cy3.5 (Abcam ab6946; 1:10), goat anti-rat Cy5 (Abcam ab6565; 1:10) and goat anti-rabbit BV480 (BD Horizon #564879; 1:10). Imaging was performed using Genomic Vision EasyScan service. Image analyses were blinded and used the Genomic Vision FiberStudio software version 2.0. For DNA combing, *n* is the number of replication events quantified from three technical replicates across two biological replicates and data/statistical analyses were performed in Microsoft Excel version 16.44 and RStudio version 1.1.442.

For telomere specific analyses of DNA replication in HCT116 super-telomerase cell lines, cells were plated as described above. Cells were incubated with 200 µM CldU (Sigma I7125) for 1 hr, rinsed with pre-warmed medium and then grown without label for 2 hr (repeated 3 additional times). Approximately 500,000 cells were embedded per plug and digested as described above. Plugs were next digested with either RsaI (NEB R0167) or HinfI (NEB R0155) restriction enzyme in 1x NEB CutSmart Buffer to degrade non-telomeric DNA. Samples were then melted, combed, baked and stored as described above. DNA was denatured in 0.5 M NaOH and 1 M NaCl for 12 min at RT. Telomeres were detected using TelG-Alexa488-conjugated PNA probe (PNA Bio F1008) with a 1:25 dilution in hybridization solution (35% formamide, 10 mM Tris-HCl pH 7.5, 0.5% blocking buffer pH 7.5 (100 mM maleic acid, 150 mM NaCl, 10% Roche blocking reagent #11096176001). Antibody staining was performed in 3% BSA in PBS-Triton (0.1%). CldU-labeled DNA was detected using primary rat anti-CldU (Abcam ab6326; 1:10), followed by secondary goat anti-rat AF555 (Invitrogen A21434; 1:10) antibodies. Imaging was performed using an EVOS FL imaging system (ThermoFisher AMF43000). Image analyses were blinded and used FIJI version 2.1.0/153.c and Adobe Photoshop version 21.2.1. For telomere specific DNA combing, *n* is the number of replication events quantified from three technical replicates across two biological replicates. Statistical analysis was performed using Microsoft Excel version 16.44.

**Clonogenic survival assay**. The number of cells plated per well is noted in each figure or figure legend. Cells were incubated for 10 to 14 days, then fixed in 10% acetic acid/10% methanol and stained with crystal violet. Colonies reaching a minimum size of 50 cells were counted manually. For all experiments, *n* is represented by biological replicates. Statistical analysis was performed using Microsoft Excel version 16.44. Plates were imaged using an Epson Expression 1680 scanner.

**Cellular senescence assay**. HCT116 cells were plated at 2,000 to 4,000 cells per well and RPE-1 cells were plated at 1000 to 1500 cells per well in 96-well plates and allowed to recover for three days. The β-galactosidase activity was measured with the 96-Well Cellular Senescence Assay Kit (Cell Biolabs CBA-231) following the manufacturer's instructions with the following modifications. Cell lysates were centrifuged in v-bottom 96-well plates rather than microcentrifuge tubes and total protein concentration was determined using Protein Assay Dye (BioRad #500-0006) following standard protocols. Plates were imaged using a VICTOR$^3$V 1420 Multilabel Counter (Perkin Elmer). The β-galactosidase activity was normalized to total protein concentration and shown as arbitrary fluorescence units. For all experiments, *n* is represented by biological replicates. Analysis and statistical tests were performed using Microsoft Excel version 16.44.

**Telomere restriction fragment (TRF) analysis**. Genomic DNA was extracted from ~1 × 10$^7$ cells using a modified version of the Gentra Puregene Cell Kit cell extraction protocol (Qiagen 158745). Integrity of genomic DNA and absence of

contaminating RNA was confirmed via 1% agarose 1x TAE gel electrophoresis. Subsequently, 30 to 40 μg of genomic DNA was digested with HinfI (NEB R0155) and RsaI (NEB R0167). For each sample, 8 to 12 μg of digested genomic DNA was resolved overnight on a 0.7% agarose 1x TBE gel. Gels were depurinated, denatured, and neutralized, followed by overnight capillary transfer to a Hybond-XL membrane (GE Healthcare RPN303S). Telomere probe was labeled using T4 polynucleotide kinase (NEB M0201) and γ-P$^{32}$-ATP (Perkin Elmer NEG035C) and purified using quick spin columns (Roche 11-273-949-001). Membranes were pre-hybridized for 1 hr with Church buffer at 55 °C, then hybridized with a γ-P$^{32}$-end-labeled telomere probe ((C$_3$TA$_2$)$_4$) in Church buffer at 55 °C overnight. Membranes were washed three times with 4x SSC and once with 4x SSC + 0.1% SDS, each for 30 min, exposed to a phosphorimaging screen, detected with a Typhoon FLA 9500 imager, and processed using FIJI version 2.1.0/153.c and Adobe Photoshop version 21.2.1. For TRF analyses using telomerase inhibitor, a 10 mM stock solution of BIBR1532 (Tocris #2981) in DMSO was prepared and diluted in appropriate growth medium to a final concentration of 10 μM as described[35].

**Telomere 2D gel analysis.** Samples were collected and prepared as described above for TRF analyses. For S1 nuclease digested samples, 20 μg of RsaI/HinfI digested DNA was digested with 100 U of S1 nuclease (Thermo Scientific #EN0321) for 45 min at room temperature. For each sample, 8 to 12 μg of digested genomic DNA was resolved overnight on a 0.5% agarose 1x TBE gel. Sample lanes were cut out, re-cast, and resolved on a 1.2% agarose 1x TBE gel. Gels were treated, transferred to Hybond-XL membrane, labeled, and imaged as described above for TRF analyses.

**Telomeric repeat amplification protocol (TRAP) assay.** TRAP assays were performed following the manufacturer's protocol for the TRAPeze Telomerase Detection Kit (EMD-Millipore S7700). Briefly, whole cell extracts were prepared using 1x CHAPS lysis buffer and stored at −80 °C. Protein concentration was measured using Protein Assay Dye (BioRad #500-0006) in comparison to a BSA (NEB B9000S) standard curve following standard protocols. TRAP reactions utilized Platinum Taq DNA Polymerase (Invitrogen 10966) and were resolved on 10% polyacrylamide 0.5x TBE gels, stained for 1 h with SYBR GOLD (Invitrogen S11494) and detected with a Typhoon FLA 9500 imager and processed using FIJI version 2.1.0/153.c and Adobe Photoshop version 21.2.1.

**Immunofluorescence analyses of telomeres.** For t-FISH asynchronous HCT116 populations were incubated for 1 hr in 0.25 μg/mL KaryoMAX colcemid (Gibco 15212-012). Cells were collected and washed in 75 mM prewarmed KCl. Cells were then fixed three times in methanol:acetic acid (3:1) by adding fixative solution dropwise with constant gentle agitation by vortex. Following fixation, cells were dropped onto microscope slides and metaphase spreads were allowed to dry overnight. Next, cells were rehydrated in 1x PBS, followed by fixation in 3.7% formaldehyde. Slides were then washed twice in 1x PBS, rinsed in ddH2O, dehydrated in an ethanol series (70%, 85%, 95%) pre-chilled to −20 °C and air dried. FISH was performed with TelC-Cy3 (PNA Bio F1002; 1:350) and CENPB-Alexa488 (F3004) warmed to 55 °C then diluted 1:300 in hybridization buffer (70% formamide, 10 mM Tris pH 7.4, 4 mM Na$_2$HPO$_4$, 0.5 mM citric acid, 0.25% TSA blocking reagent (Perkin Elmer FP1012), and 1.25 mM MgCl$_2$) preheated to 80 °C. Slides were denatured with probe at 80 °C, then allowed to incubate at room temperature in a humid chamber for 2 hr. Next, slides were washed twice in PNA wash A (70% formamide, 0.1% BSA, 10 mM Tris pH 7.2) and three times in PNA wash B (100 mM Tris pH 7.2, 150 mM NaCl, 0.1% Tween-20). The second PNA wash B contained DAPI (Life Technologies D1306) at a 1:1000 concentration. Slides were then dehydrated and dried as described above prior to mounting. Slides were blinded prior to imaging and captured using a Zeiss Spinning Disk confocal microscope. Image analyses were blinded and used FIJI version 2.1.0/153.c and Adobe Photoshop version 21.2.1. Statistical analysis was performed using Microsoft Excel version 16.44.

For analyses of telomere sister chromatid exchange, HCT116 ST cells were cultured in the presence of BrdU:BrdC (final concentration of 7.5 mM BrdU (MP Biomedicals 100166) and 2.5 mM BrdC (Sigma B5002)) for 12 hr prior to harvesting. KaryoMAX colcemid (Gibco 15212-012) was added at a concentration of 0.1 μg/mL during the last two hr. CO-FISH was performed[73] using a TelC-Alexa488-conjugated PNA probe (PNA Bio F1004; 1:1,000) followed by a TelG-Cy3-conjugated PNA probe (PNA Bio F1006; 1:1,000), then washed and mounted following the staining protocol described above for t-FISH. Images were captured using a Zeiss Spinning Disk confocal microscope. Image analyses were blinded and used FIJI version 2.1.0/153.c and Adobe Photoshop version 21.2.1. Statistical analysis was performed using Microsoft Excel version 16.44.

**Statistics and reproducibility.** Statistical details of experiments, including what n represents, are described in the corresponding Methods Details section for each technique and/or in each figure legend. Further explanation of what n represents, and the value of n is defined in each figure legend. Dispersion and precision measures (e.g., average, standard deviation (SD)), as well as how significance was defined is indicated in each corresponding figure legend. The statistical test used for all experiments was an unpaired, two-tailed student's t-test, except Fig. 2g–l, which

used an unpaired, two-tailed Mann-Whitney Ranked Sum Test. The software used for statistical analysis of each type of experiment is indicated in the corresponding Method Details section.

Experimental findings were reproduced in a minimum of two biological replicates and typically in multiple independent, clonal cell lines. All results were reproducible. The only exceptions were the "+ telomerase inhibitor" samples on the TRF gel in Fig. 6b. This represents the only reverted MCM10 mutant cell line that arose during growth in telomerase inhibitor. We continued to propagate in the presence of telomerase inhibitor for 67 population doublings and analyzed 9 separate timepoints, all of which showed a reproducible phenotype that was consistent with all other analyses in the paper.

**Reporting summary.** Further information on research design is available in the Nature Research Reporting Summary linked to this article.

## Data availability
The authors declare that all data related to the findings of this study are available within the article and supplementary information or are available from the corresponding author upon reasonable request. Source data are provided with this paper. The Protein Data Bank accession codes for the *Xenopus laevis* MCM10-ID are 3EBE[16] and 3H15[74]).

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

## Acknowledgements

We thank members of the Bielinsky laboratory for helpful discussions. We also thank L. Harrington for generously sharing reagents and R. Rebbeck for help with CD experiments. This work was supported by NIH grants GM074917 (A.K.B.), GM134681 (A.K.B.), GM083024 (J.G.C.), GM102413 (J.G.C.), CA190492 (E.A.H.), GM118047 (H.A.) and T32-CA009138 (R.M.B., W.L., C.B.R.), by NSF Fellowship DGE-1144081 (J.P.M.), a UNC Dissertation Completion Fellowship (J.P.M.), NIH National Center for Advancing Translational Sciences grants TL1R002493 and UL1TR002494 (M.M.S.) and the ARCS Foundation (C.B.R.). We wish to acknowledge the University of Minnesota Flow Cytometry Resource, the University of Minnesota Imaging Centers, the University of Minnesota Genomics Center, and the Masonic Cancer Center Cancer Genomics Shared Resource supported by P30 CA077598. The University of North Carolina Flow Cytometry Core Facility is supported in part by P30 CA016086. Exome sequencing was supported in part by the National Institute for Health Research Oxford Biomedical

Research Centre Program. Genome sequencing was conducted as part of an independent research program commissioned by the Health Innovation Challenge Fund (R6-388/WT 100127), a parallel funding partnership between the Wellcome Trust and the Department of Health. The views expressed in this publication are those of the authors and not necessarily those of the Wellcome Trust or the Department of Health.

## Author contributions

R.M.B. designed and performed experiments, analyzed data, prepared the figures, and co-wrote the manuscript. W.L., M.M.S., J.P.M., L.Y., M.K.O., J.T., A.T.P., J.C.T., H.D., J.C., E.O., H.W., and E.B. performed experiments and analyzed data. G.S. helped with data interpretation. L.W., C.B.R., D.B., J.H., and A.H. performed experiments. A.K.B. designed experiments, supervised the study, and co-wrote the manuscript. All authors including E.A.H., E.M.M., J.S.O., H.A., and J.G.C. reviewed and edited the manuscript.

## Competing interests

The authors declare no competing interests.
