## [Peer Review File · Nature Communications]

REVIEWER COMMENTS

Reviewer #1 (Remarks to the Author):

The authors have studied the role of the DNA replication helicase MCM10 in DNA and telomere replication. Recent approaches, as employed here, to genetically engineer human cells to contain only one functional allele, have proven invaluable to illuminate the dosage-sensitive role of proteins in cell function and human disease. MCM10 is no exception; and these authors describe several key findings. First, they show that MCM10 is haploinsufficient (in cell lines), and its absence is more acutely manifested in a colon cancer cell line (HCT116) than in primary immortalized cells (RPE-1). Secondly, they describe two functions for MCM10 that are illuminated in the heterozygous lines: (i) MCM10 heterozygosity leads to decreased origin firing and cell cycle progression; which may be the first evidence of a dosage-sensitive role for MCM10 in DNA replication, and (ii) an unexpected and specific role for MCM10 in telomere maintenance that is not observed in cells heterozygous for other DNA replication machinery components such as CDC45 and MCM4. There is a wealth of data in this study; the experiments are well conducted and clearly presented. The discussion was thorough and incisive. This study will be of broad interest to readers due to several new insights provided in how MCM10 function impacts human disease, DNA replication, and telomeres. There were only minor comments that can be addressed via text modifications, below:

The telomere replication and length regulation defects are highly suggestive of a potential mechanism for the human diseases NKD and RCM in which the MCM10 variants are bi-allelic. One minor point to keep in mind is that a definitive demonstration of the role of telomere maintenance would be pending some demonstration such as separation-of-function alleles of MCM10 that affect telomere replication without affecting DNA replication origin firing. Notwithstanding this small caveat, the authors are justified in the suggestions and conclusions that they draw here.

The telomere phenotype is compelling and is extensively characterized in both cell lines. The authors also measured beta-galactosidase activity in MCM10^{+/-} cells, which was elevated. They state that this data corroborates the telomere defect (which it does) but it should be noted that elevated beta-galactosidase is also observed in other types of stress-induced senescence that is not necessarily telomere-induced (e.g. oncogene-induced senescence or even in response to DNA replicative stress; see doi: 10.1002/2211-5463.12632).

The reversion in MCM10 alleles in long-term propagated cultures (>75 PDL) is fascinating and is extensively characterized. Would the authors like to speculate on whether such reversion could happen in patients? It is slightly reminiscent of a Cech lab study which found that expression of a TPP1 variant is lost in cells simultaneously treated with the telomerase inhibitor BIBR1532 (doi: 10.1074/jbc.M113.518175). This study, like the Cech study, also found interesting effects on telomere length dynamics that were telomerase-dependent. The telomere dynamics in the reverted cells might be suggestive of a specific synthetic sick/lethal (SSL) interaction between the MCM10 variants and telomere replication (as opposed to DNA replication in general), although the authors found that TERT overexpression does not fully rescue all MCM10^{+/-} phenotypes.

Figure 3f. Please specify the PDL range of the "late passage" samples. Please provide more details on lanes 1, 2, 3. Are these different isolates each propagated to late passage, i.e. the same as in 3d? Arrows at the right would help clarify what each band represents.

Figure 3g. Please add DNA markers at the side of the genotyping PCR gel (and arrows to indicate the nature of each band). One could do the same for Fig 5a.

Reviewer #2 (Remarks to the Author):

In their manuscript, Baxley et. al. study the function of MCM10 in telomere length control and DNA replication. This is an interesting question because although much is already known about the specific functions of MCM10 in normal DNA replication and oncogenic transformation, little is

known about other roles of MCM10 in human disease, especially regarding inactivating mutations. Specifically, the authors identified two familial instances of compound heterozygous mutations in MCM10 which are associated with natural killer cell deficiency and restrictive cardiomyopathy with thymic and splenic hypoplasia. To better understand how inactivating mutations in MCM10 drive these diseases, the authors genetically engineer a series of human cell lines with mutations in MCM10. In these cells they find a wide spectrum of phenotypes, including telomere shortening and proliferative defects, which seem to be dependent on both the specific MCM10 mutation introduced and the parental cell line. Due to the wide range of documented phenotypes and cell-line specific idiosyncrasies, I am not convinced at this point that the experiments presented allow any robust conclusions warranting publication. My concerns are summarized below:

Major Concerns:

Cell lines are not properly described and/or characterized:

The generation of the mutant exon 14 HCT116 cell line is inadequately described. It seems that the exon 14 mutations were introduced via a sequential integration of two loxP sites followed by deletion of exon 14 through cre recombination. This process seems to have resulted in a compound heterozygous cell line wherein one allele retains exon 14 and a single loxP site while the other contains an exon 14 deletion and a loxP "scar". As a consequence, there is no true wild-type allele and the remaining exon14-loxP allele is uncharacterized. This editing process raises two main concerns:

All clones for the exon 14 cell lines have a recent common single cell precursor generated during loxP integration. Therefore these cells cannot be considered truly independent clones and should not be compared to bulk wild-type cells.

The presumptive wild-type allele is not characterized. It seems quite possible that the loxP site interferes with MCM10 production from this allele. This could explain in part why the authors see such a pronounced phenotype for this cell line, whereas exon 3 mutants in RPE-1 show a dramatically reduced phenotype.

Concern about the citation #12 as a preprint: this is foundational data for this manuscript, though it is not yet peer reviewed and under a different senior author. The requires clarification and the technical data has to become part of this paper as it otherwise cannot be evaluated.

The phenotype of the parent carrying the 236.delG mutation directly contradicts claims of haploinsufficiency. When paired with the data in figure 4 demonstrating that RPE-1 cells show much milder phenotypes, this indicates that the phenotype of the HCT116 exon 14 mutants likely results from mutation-specific or cell line-specific idiosyncrasies and should be treated as an exception rather than the rule.

"Reversion" of two edited alleles to two wild-type unedited alleles cannot be explained by known molecular mechanisms and no mechanism for "reversion" is explored or even proposed.

Additional Concerns:

The authors claim to explore patient mutations, but no true patient mutations were investigated. Missense mutations were ignored in the paper despite compound heterozygosity being essential for the human phenotype. No genotyping data for generated cell lines is shown to indicate similarity to sequenced disease-associated alleles (except for PCR genotyping of the "revertants" which have an unexplained non-specific band only present in the mutant cells).

Division of apoptotic cells into early, late, and dead cells is immaterial and the method of classification is not explained (eg. Figure 1i).

Disparities between some TRF gels indicate loading inconsistencies: In figure 3a, telomeres appear to shorten by <50% over the course of 38 population doublings (PDs), however the signal intensity of the lanes appears to drop by more than 50%. Underloading late PD wells would create the illusion of more dramatic telomere shortening. This is supported by figure 3g wherein the same cell lines appear to shorten by less than 1kb in 31 PDs prior to "reversion".

Broad distribution of telomere lengths in late PD HCT116 MCM10+/- (Figure 3d) indicate strong effects of subcloning/independent cultures which muddy the interpretation of telomere shortening. Specifically, it is still not clear what drives the proliferation phenotype in the cells. It can not be telomere length as the super telomerase mutants still grow poorly (figure 5f), as do the RPE-1 cells (figure 4b). Yet, the data indicate that "reversion" occurs when telomeres are short. Why would that happen if the selective pressure is not coming from short telomeres?

It is also surprising that late passage HCT116 MCM10+/- cells have an elevated (but not increased compared to early passage) frequency of signal-free telomere ends (SFEs) (supplementary figure

2g). If “reversion” occurs and telomere length is restored comparable to wt cells, one would expect that the frequency of SFEs to similarly be restored to wt levels. Otherwise, if telomere shortening continues, one would expect elevated SFEs. This apparent contradiction is not addressed. Figures 6a-c explore the inability of super telomerase cells to fully replicate telomeres. However the resolution of telomere lengths in figure 5c is not sufficient to confirm that WTST3, 8ST5 and 14ST4 have comparable telomere lengths. If anything, WTST3 appears to have slightly shorter telomeres, which could explain its increased ability to fully replicate the telomere within a given window of replication, independent of MCM10 activity. The interpretation of the Mcm10/Mus81 genetic interaction is not satisfactory. While the mutations appear to be additive, the analysis does not clearly indicate a synergistic effect.

Reviewer #3 (Remarks to the Author):

In this manuscript, Baxley et al report the novel association of compound heterozygous MCM10 loss-of-function (LOF) and hypomorphic variants with a severe fetal phenotype of restrictive cardiomyopathy with lymphoreticular hypoplasia. They propose underlying these clinical features, as well as NK cell deficiency associated with a different set of MCM10 variants described in an accompanying manuscript, is telomere erosion due to a critical role for MCM10 in the restart of stalled replication forks within telomeres with secondary effects on cell proliferation and survival. They present a rigorous body of data analyzing heterozygous MCM10 +/- cells, demonstrating marked effects on telomeres in the HCT116 cancer cell line, with the striking emergence of MCM10 +/- revertants, in hTERT-immortalized RPE-1 cells, particularly in the presence of telomerase inhibition, and in super telomerase HCT116 cells, reflected by accumulation of t complex DNA. They show a role for Mus81 in maintaining telomeres in the context of MCM10 +/- and a lack of a role for replisome proteins Cdc45 and Mcm4. The manuscript makes important contributions to the telomere field although it leaves open the question as why the clinical phenotypes do not overlap more significantly with those associated with mutations in the numerous genes identified to date in the telomere biology disorders. The manuscript would be strengthened by the authors addressing the following points:

Major points

1. The cell line data reported here demonstrate MCM10 is genetically haploinsufficient, with striking telomere-related phenotypes and impacts on cell proliferation and survival. GnomAD data, however, suggests that there is not an intolerance of loss-of-function MCM10 variants (pLI 0, o/e 0.84). Additionally, there is an absence of clinical phenotypes in the mother of the family described here or the mother described in the Mace et al, accompanying manuscript, who carried loss-of-function (LOF) variants. Analysis of telomere length (preferably by telomere flow FISH including CD56+ cells) of the parents and offspring without biallelic MCM10 variants would help clarify if MCM10 haploinsufficiency is also observed in primary human cells.
2. The description c.764+5G>A, p.D198GfsTer10 does not make sense to me. First, the coding DNA position 764 does not correspond to the third nucleotide of the D198 codon. Also, p.D198GfsTer10 would not be correct given the predicted impact on RNA splicing. See <https://varnomen.hgvs.org/recommendations/RNA/variant/splicing/> for standard splicing variant nomenclature.
3. Pg 23, line 508, I disagree with the statement that cardiomyopathy is a well-recognized feature of dyskeratosis congenita. The 2008 Basel-Vanagaite report highlighted the novelty of cardiac abnormality in patients with DC and proposed this be studied in a larger cohort (which I don't believe has been done). The reviews by Bessler et al, and Savage similar indicate cardiac abnormalities are rare.
4. Figures 6a-c demonstrate impaired telomere replication with increased percentage of unreplicated and partially replicated telomeres in the MCM10 +/- ST lines. While the 2D gels clearly show accumulated t-complex DNA, the TRFs analysis in Figure 5c does not show accumulation of short TRFs or increased telomere length heterogeneity. Might this be due to differences in S phase distribution or transit time?
5. If MCM10 contributes to telomere length maintenance by stimulating replication restart within duplex telomeric tracts, then the requirement or impact of MCM10 deficiency might differ in cells with long versus short telomeres with greater impact on those with longer telomeres. Can attrition

rates be determined in the various cell lines, and possibly in the ST lines following telomerase inhibition?

6. Pg 13, lines 292-3, the statement that Cdc45 levels remained normal in MCM10+/- cells lines is not supported by the data which is variable with levels as low as 0.6X WT (Supplemental fig. 3).

Minor points

7. The word mutation(s) should be replaced by variant(s) (see Richards et al, Standards and guidelines for the interpretation of sequence variants: a joint consensus recommendation of the American College of Medical Genetics and Genomics and the Association for Molecular Pathology, Genet Med 2015).

8. Not all of the research subjects were patients e.g., parents and unaffected siblings.

9. Pg 3, line 60, CM should be RCM.

10. Pg 5, lines 104-106, given a strong telomere phenotype was observed MCM10 +/-, it suggests that MUS81 is unable to prevent fork collapse.

11. Pg 6, line 125 and Pg 8, line 163, c.236delG is not a nonsense variant.

12. Pg 10, the description of numbers of novel karyotypes but reference to a table that lists specific chromosomal aberrations (not karyotypes) is confusing.

13. Pg 11, line 23, highlight that the other cells were those with exon 14 targeted, as it is not stated. Indeed, it might be good to state early that for most experiments clones obtained after exon 14 targeting were analyzed.

14. Pg 13, line 294, Fig. 1d should be Fig. 1f.

15. Pg 20, line 437, the RPE-1 cells are TERT immortalized, correct? This needs to be stated.

16. Pg 23, line 520, if amino acids 198-255 are deleted in the splice site variant (pg 7, line 138), then the number of deleted amino acids should be 258 not 255.

17. Pg 25, line 553, spell-out GM-MDT.

18. Supplemental Figure 1b legend, it should be explicitly stated that sequencing is of cDNA, presumably cloned for the father.

19. How was the structural model of human Mcm10-ID in Fig. 1c generated or is it the Xenopus structure with the position of the cognate R426C residue and those that would be deleted if human exon 6 was skipped shown?

20. Add kB above ladder numbers in Fig. 5c left and right telomere Southern blots.

21. DC is now more frequently used than DKC to abbreviate dyskeratosis congenita to limit assumed reference to DKC1.

Reviewer comments are listed as black text. Our responses are listed below each comment in *blue italicized text*. Our references to page and line number refer to the revised version of our manuscript.

In addition to changes made in response to the reviewers, two changes have been made in compliance with *Nature Communications* editorial and reporting policies.

- 1) All figures have been modified such that individual data points are shown when possible, and always for $n \leq 10$. In each figure legend, we state whether all data points or the average of replicate data points is indicated.
- 2) We have added Supplemental Fig 2d, which outlines the gating strategy used for flow cytometry analyses of apoptosis.

Reviewer #1 (Remarks to the Author):

The authors have studied the role of the DNA replication helicase MCM10 in DNA and telomere replication. Recent approaches, as employed here, to genetically engineer human cells to contain only one functional allele, have proven invaluable to illuminate the dosage-sensitive role of proteins in cell function and human disease. MCM10 is no exception; and these authors describe several key findings. First, they show that MCM10 is haploinsufficient (in cell lines), and its absence is more acutely manifested in a colon cancer cell line (HCT116) than in primary immortalized cells (RPE-1). Secondly, they describe two functions for MCM10 that are illuminated in the heterozygous lines: (i) MCM10 heterozygosity leads to decreased origin firing and cell cycle progression; which may be the first evidence of a dosage-sensitive role for MCM10 in DNA replication, and (ii) an unexpected and specific role for MCM10 in telomere maintenance that is not observed in cells heterozygous for other DNA replication machinery components such as CDC45 and MCM4. There is a wealth of data in this study; the experiments are well conducted and clearly presented. The discussion was thorough and incisive. This study will be of broad interest to readers due to several new insights provided in how MCM10 function impacts human disease, DNA replication, and telomeres. There were only minor comments that can be addressed via text modifications, below:

We thank the Reviewer for the positive feedback.

1) The telomere replication and length regulation defects are highly suggestive of a potential mechanism for the human diseases NKD and RCM in which the MCM10 variants are bi-allelic. One minor point to keep in mind is that a definitive demonstration of the role of telomere maintenance would be pending some demonstration such as separation-of-function alleles of MCM10 that affect telomere replication without affecting DNA replication origin firing. Notwithstanding this small caveat, the authors are justified in the suggestions and conclusions that they draw here.

We agree that our data do not provide definitive evidence that MCM10-dependent telomere maintenance is causative for NKD and RCM. We added to our discussion on page 26, lines 579 to 582 that “Future analysis of different human cell lineages and/or separation-of-function MCM10 mutations would be required to demonstrate that MCM10-dependent telomere maintenance defects are causative in these human diseases.”

2) The telomere phenotype is compelling and is extensively characterized in both cell lines. The authors also measured beta-galactosidase activity in MCM10^{+/-} cells, which was elevated. They state that this data corroborates the telomere defect (which it does) but it should be noted that elevated beta-galactosidase is also observed in other types of stress-induced senescence that is not necessarily telomere-induced (e.g. oncogene-induced senescence or even in response to DNA replicative stress; see doi:10.1002/2211-5463.12632).

We agree with this statement and have modified the text on page 12, lines 258 to 261 to reflect this point as follows: “We observed significantly higher activity in mutant cell extracts, further corroborating the telomere maintenance defect, although this does not rule out other sources of stress-induced senescence pathway activation.”

3) The reversion in MCM10 alleles in long-term propagated cultures (>75 PDL) is fascinating and is extensively characterized. Would the authors like to speculate on whether such reversion could happen in patients? It is slightly reminiscent of a Cech lab study which found that expression of a TPP1 variant is lost in cells simultaneously treated with the telomerase inhibitor BIBR1532 (doi: 10.1074/jbc.M113.518175). This study, like the Cech study, also found interesting effects on telomere length dynamics that were telomerase-dependent. The telomere dynamics in the reverted cells might be suggestive of a specific synthetic sick/lethal (SSL) interaction between the MCM10 variants and telomere replication (as opposed to DNA replication in general), although the authors found that TERT overexpression does not fully rescue all MCM10^{+/-} phenotypes.

We speculate that the reversion is facilitated by the loxP sites (please see response to Reviewer 2, major concern #4), because we have only observed reversion in rAAV-altered HCT116 cell lines. We would therefore predict that it might not happen in patients.

The Reviewer makes an interesting suggestion about synthetic interactions between MCM10 variants and genes specific for telomere replication. This is something we will systematically pursue in the future. As the Reviewer stated, increased telomerase activity does not rescue the replication defects in MCM10 mutants. Based on this, we do not favor the interpretation that a specific synthetic interaction exists, but this requires further investigation.

4) Figure 3f. Please specify the PDL range of the “late passage” samples. Please provide more details on lanes 1, 2, 3. Are these different isolates each propagated to late passage, i.e. the same as in 3d? Arrows at the right would help clarify what each band represents.

The genomic DNA for the populations analyzed in the TRF for Fig. 3d is the same as used for the PCR in Fig. 3f. We specify in the text that these were six independent populations at >75 PDs (page 12, line 268). In the revised manuscript, we also added this information to the figure legend and included it in the figure itself. Furthermore, we modified the organization and schematics of the PCR products in Fig. 3f to address this comment.

5) Figure 3g. Please add DNA markers at the side of the genotyping PCR gel (and arrows to indicate the nature of each band). One could do the same for Fig 5a.

We changed both figure panels according to the Reviewer's suggestion.

Reviewer #2 (Remarks to the Author):

In their manuscript, Baxley et. al. study the function of MCM10 in telomere length control and DNA replication. This is an interesting question because although much is already known about the specific functions of MCM10 in normal DNA replication and oncogenic transformation, little is known about other roles of MCM10 in human disease, especially regarding inactivating mutations. Specifically, the authors identified two familial instances of compound heterozygous mutations in MCM10 which are associated with natural killer cell deficiency and restrictive cardiomyopathy with thymic and splenic hypoplasia. To better understand how inactivating mutations in MCM10 drive these diseases, the authors genetically engineer a series of human cell lines with mutations in MCM10. In these cells they find a wide spectrum of phenotypes, including telomere shortening and proliferative defects, which seem to be dependent on both the specific MCM10 mutation introduced and the parental cell line. Due to the wide range of documented phenotypes and cell-line specific idiosyncrasies, I am not convinced at this point that the experiments presented allow any robust conclusions warranting publication. My concerns are summarized below:

We thank the Reviewer for finding this work interesting. We are unsure what the Reviewer means by “cell-line specific idiosyncrasies”, as we document similar phenotypes (diminished proliferation and telomere shortening) in both cell lines (HCT116 and hTERT RPE-1) we used in this study. We hope that our answers to the specific questions below will clarify the concerns this Reviewer raised.

Major Concerns:

1) Cell lines are not properly described and/or characterized:

The generation of the mutant exon 14 HCT116 cell line is inadequately described. It seems that the exon 14 mutations were introduced via a sequential integration of two loxP sites followed by deletion of exon 14 through Cre recombination. This process seems to have resulted in a compound heterozygous cell line wherein one allele retains exon 14 and a single loxP site while the other contains an exon 14 deletion and a loxP “scar”.

This is an accurate description.

As a consequence, there is no true wild-type allele and the remaining exon14-loxP allele is uncharacterized.

It is correct that there is no “true” wild-type allele in the rAAV generated mutants. This is why we also utilized CRISPR/Cas9 to generate an independent set of heterozygous HCT116 cell lines. More importantly, however, we present extensive characterization of the remaining allele that retains exon 14 and a single loxP site in multiple clonal cell lines:

- 1) *We demonstrate that the loxP allele expresses approximately 50% of MCM10 protein in comparison to wild-type cells (ranging from 50 to 70% in Fig. 1e “exon 14” panel).*
- 2) *We demonstrate that the level of MCM10 expression from this allele is similar to the amount expressed from the only remaining wild-type allele in independent heterozygous HCT116 cell lines derived from the same parental line (ranging from 40 to 60% in Fig. 1e “exon 3” panel). We believe that based on this data it is fair to conclude that the loxP-containing allele behaves like a wild-type allele.*

This editing process raises two main concerns: all clones for the exon 14 cell lines have a recent common single cell precursor generated during loxP integration. Therefore these cells cannot be considered truly independent clones and should not be compared to bulk wild-type cells. The presumptive wild-type allele is not characterized. It seems quite possible that the loxP site interferes with MCM10 production from this allele. This could explain in part why the authors see such a pronounced phenotype for this cell line, whereas exon 3 mutants in RPE-1 show a dramatically reduced phenotype.

We respectfully disagree. A study by the Hendrickson laboratory demonstrated that the incorporation of “mismatches” is highly dependent on their location within the homology arms relative to the antibiotic resistance marker (Kan et. al., 2014; DOI: 10.1371/journal.pgen.1004251). We assessed the probability for the two lox-P sites to be incorporated into the flanking regions of exon 14 based on the above-mentioned study, where the probability of integration decreases linearly (~1% per 10 bp) with distance from the antibiotic resistance marker. We determined that the integration probabilities for the upstream and downstream loxP sites are ~ 0.45 and 0.90, respectively. We screened 103 neomycin resistant colonies, 14 of which tested positive by PCR for locus-specific targeting. We randomly selected 4 clones for further genotyping. Of these, 2 had incorporated both loxP sites, and 2 only had the downstream loxP site, consistent with the predicted probabilities (although we concede that these are small numbers). The clones that were lacking the loxP site upstream of exon 14 but contained a floxed NEO gene easily excised the NEO marker after transient introduction of Cre-recombinase. We would agree with the Reviewer – that all clones might derive from a common cell precursor – if the probability for loxP site integration were close to 100% for the upstream loxP site, but it isn't. We have included the above information in the methods section “Cell line generation using rAAV” beginning on page 31, line 697. Even if we assume that the Reviewer is correct and that all four clones were derived from a common precursor, we have also demonstrated that completely independently-derived heterozygous cell lines behave very similarly (targeted MCM10 exon 3; Fig. 1e and f; Supplemental Fig. 3c).

The second concern deals with the fact that the loxP site might interfere with MCM10 production. Based on the sequencing results that demonstrate that exon 14 retains wild-type coding and splicing information, we deem this possibility unlikely. We have included this information in the revised manuscript in a new supplemental figure (Supplemental Fig. 2). As mentioned above, we carefully quantified expression levels of MCM10, the production of which was cut roughly in half, as expected for a heterozygous cell line. Importantly, we observed a similar expression range for CRISPR-engineered HCT116 and RPE-1 mutants (that don't carry

a loxP site). Interestingly, in all instances – without exception – the proliferation behavior of a particular cell line correlated with MCM10 expression levels. These results argue that the loxP site has no effect on MCM10 expression and that the loxP allele behaves like a wild-type allele. Consequently, the differences in phenotypes between the MCM10-deficient HCT116 and RPE-1 cell lines can't be due to differences in expression levels of MCM10. We favor the idea that the differences are due to the fact that one cell line is transformed and the other is not.

2) Concern about citation #12 as a preprint: this is foundational data for this manuscript, though it is not yet peer reviewed and under a different senior author. This requires clarification and the technical data has to become part of this paper as it otherwise cannot be evaluated.

The paper referenced in citation #12 – Mace, E. M. et al. Human NK cell deficiency as a result of biallelic mutations in MCM10. bioRxiv, 825554, DOI:10.1101/825554 (2019) – has recently been published in The Journal of Clinical Investigation (DOI:10.1172/JCI134966). We have updated our citation of this paper in the manuscript. The characterization presented in this paper was supervised by Dr. Jordan Orange. Mace et al. contains a completely different set of experiments without any overlap with our manuscript. We concentrated on the telomere and replication phenotypes, whereas Dr. Orange's group treated the NK cell patient and sequenced the DNA of the patient's family.

3) The phenotype of the parent carrying the 236.delG mutation directly contradicts claims of haploinsufficiency. When paired with the data in figure 4 demonstrating that RPE-1 cells show much milder phenotypes, this indicates that the phenotype of the HCT116 exon 14 mutants likely results from mutation-specific or cell line-specific idiosyncrasies and should be treated as an exception rather than the rule.

We carefully differentiate between organismal and cellular haploinsufficiency, and do not claim that the 236.delG mutation causes haploinsufficiency on an organismal level. In fact, we agree with the Reviewer, stating in the manuscript that “The absence of pathology in family members carrying a single MCM10 variant demonstrates that mono-allelic null mutations are not haploinsufficient in human beings.” (page 25, lines 559 to 560).

What we describe are cellular phenotypes that may directly contribute to the diseases diagnosed in the patients. Based on the observation that clinical phenotypes are caused only when the parental mutations are combined, we propose that – on the organismal level – there must be a cell-type specific threshold of MCM10 expression. Nevertheless, it is surprising that the deletion of a single allele of MCM10 causes proliferation and telomere maintenance defects, especially since mice heterozygous for MCM10 were viable, fertile and healthy (Lim et. al., 2011; DOI: 10.1016/j.bbamcr.2011.05.012). It is true that HCT116 cells show a more severe phenotype than RPE-1 cells, however, RPE-1 cells also display genetic haploinsufficiency, as they proliferate slower than wild-type cells (Fig. 4b), have increased levels of apoptosis (Fig. 4c), have fewer active replication forks (Fig. 4i), and exhibit defects in telomere length maintenance (Fig. 4m-o). These are clear commonalities. The fact that RPE-1 mutant cells exhibit milder phenotypes than HCT116 mutant cells does not negate data meeting the basic criteria for MCM10 haploinsufficiency in both cell types.

4) “Reversion” of two edited alleles to two wild-type unedited alleles cannot be explained by known molecular mechanisms and no mechanism for “reversion” is explored or even proposed.

The genetic reversion is indeed a very interesting point. We are happy to include a proposal of the mechanism underlying the reversion. We observed two types of reversion, as illustrated in Fig. 3f. In one scenario, the reverted cell line retained the loxP sites downstream of exon 14. In the other scenario, the reverted line lost the loxP site. Reversion could be explained by a break-induced replication (BIR)-like mechanism that we would describe as follows on page 14, lines 294 to 298: “We hypothesize that reversion occurred through a homology-dependent recombination mechanism (Supplemental Fig. 3h), such as break-induced replication (BIR)³⁰, following the random introduction of a DNA break at the MCM10 locus, caused by the underlying increase in genomic instability in the mutant cell lines.”

We speculate that the repetitive sequence of the loxP sites may play a role in stimulating reversion, either by increasing the propensity for replication fork stalling and collapse into a double-stranded break, or by facilitating recombination at the locus, even in the absence of Cre-recombinase. If the break occurs upstream of the loxP scar (left side of Supp. Fig. 3h, see below), the left side of the break could theoretically invade the homologous chromosome upstream of exon 14 due to sequence homology. BIR replicates extended DNA regions by displacement DNA synthesis and could easily copy exon 14 and the downstream loxP site of the homolog. Importantly, BIR is known to occur in “two flavors”, either driven by long regions of homology (RAD51-dependent), or very short regions, often just a few nucleotides, of microhomology (RAD52-dependent). The second scenario might be the result of microhomology-mediated BIR (right side of Supp. Fig. 3h, see below) where the break occurs downstream of the loxP scar and the right double-stranded end invades the homolog immediately upstream of the loxP site and copies exon 14 in the opposite direction. We have depicted these events in a schematic to be included in Supplemental Fig. 3h (and see below). We assume that these are very rare events happening in single cells. Because the reversion provides a selective advantage against the MCM10^{+/-} population, it allows the reverted cell to expand and slowly “take over” the population. This is consistent with the changes we monitored by PCR genotyping over time (Fig. 3g; Fig. 5a; Supplemental Fig. 3g).”

Supp. Fig. 3h:

Additional Concerns:

1) The authors claim to explore patient mutations, but no true patient mutations were investigated.

We spent multiple years (and lots of resources) attempting to create true compound heterozygous cell lines using CRISPR editing. We were unable to obtain such cells, even after outsourcing the project to the Genome Editing Shared Resource in the Masonic Cancer Center at the University of Minnesota. What we did instead was to introduce the exact exon 13 mutation (only viable in a heterozygous configuration), and exact exon 10 mutation (in a homozygous configuration) that was discovered in the NK cell patient. Both of these mutants have a proliferation defect (Mace, et al., 2020, DOI:10.1172/JCI134966.) and telomere shortening (Fig. 4o). Moreover, the exon 14 deletion (if expressed) very closely resembles the exon 13 mutation (it has precisely one additional amino acid, Supplemental Fig. 1d). Lastly, we also targeted exon 3 in both HCT116 and RPE-1 cells to mimic the exon 3 variant in the RCM patients. We have improved Supplemental Fig. 1d and added Supplemental Fig. 2a-c to clarify which mutations we generated in tissue culture cells.

Although we did not generate a cell line carrying the splice site variant, we performed in vitro experiments to understand the outcome of the most functional consequence of this mutation. Extensive structural analysis suggested that any expressed protein variant is prone to aggregation (Fig. 1c,d, Supplemental Fig. 1e,f).

2) Missense mutations were ignored in the paper despite compound heterozygosity being essential for the human phenotype.

We respectfully disagree with this statement. In Fig. 4o, we show that the exon 10 missense mutation causes a short telomere phenotype in RPE-1 cells.

3) No genotyping data for generated cell lines is shown to indicate similarity to sequenced disease-associated alleles (except for PCR genotyping of the “revertants” which have an unexplained non-specific band only present in the mutant cells).

We have included the requested data. In Supplemental Fig. 2, we have added sequencing data for HCT116 and RPE-1 exon 3 mutants to show the similarity with the c.236delG mutation, and indicated the location of exon 14 and exon 3 cell line mutations on the schematic of MCM10 in Supplemental Fig. 1d to more clearly show the similarity with disease-associated alleles. With respect to the very faint non-specific PCR band, we attempted to sequence the band, but were never able to get enough material.

4) Division of apoptotic cells into early, late, and dead cells is immaterial and the method of classification is not explained (e.g. Figure 1i).

To assess the apoptotic proportion of the examined cell populations, we employed a commonly used commercial annexin V assay. The division is based on whether cells are positive for both propidium iodide and annexin V staining, positive for only one, or negative for both. This allows the identification of both early and late apoptotic cells, as well as cells that may have activated a different cell death mechanism. We added representative flow sort data in

Supplemental Fig. 2d for clarity and added additional text to the figure legend to read “Average percentage of each population represented by early apoptotic, late apoptotic or dead cells as detected using a combination of propidium iodide and annexin V staining.”

5) Disparities between some TRF gels indicate loading inconsistencies: in figure 3a, telomeres appear to shorten by <50% over the course of 38 population doublings (PDs), however the signal intensity of the lanes appears to drop by more than 50%. Underloading late PD wells would create the illusion of more dramatic telomere shortening. This is supported by figure 3g wherein the same cell lines appear to shorten by less than 1 kb in 31 PDs prior to “reversion”. Broad distribution of telomere lengths in late PD HCT116 MCM10+/- (Figure 3d) indicate strong effects of subcloning/independent cultures which muddy the interpretation of telomere shortening.

We performed TRF assays following well established protocols and with the utmost care to avoid loading inconsistencies. Genomic DNA is harvested from $\sim 1 \times 10^7$ cells per sample. DNA integrity and lack of RNA contamination is confirmed via gel electrophoresis prior to any further analysis. DNA is digested with two restriction enzymes (HinfI and RsaI) to degrade genomic DNA and enrich for telomeric fragments. Following phenol-chloroform purification of telomeric fragments, sample concentration is quantified by spectrophotometry. Based on this, identical amounts of each telomere-enriched sample are loaded in each lane. It is true that overall signal intensity may vary between lanes due to differences in average telomere length (shorter telomeres are usually fainter, because they capture less probe and resolve more efficiently on the gel). This is, however, not caused by “underloading”. The overall size distribution and sample resolution is not affected by the amount of DNA that is loaded. In other words, the amount of DNA loaded does not alter the molecular weight distribution of telomeres in a sample. Thus, the region of peak signal intensity within each sample will not change. This is why on each lane of every TRF assay we have indicated the location of peak signal intensity with a yellow oval.

We agree that the rate of telomere erosion observed in MCM10 mutant cell lines is variable between independent cultures. This variability is dramatically highlighted by the observation that some (but not all!) cultures undergo reversion of the MCM10 mutation, causing telomere lengthening (Fig. 3g, Fig. 5b, Supplemental Fig. 3g). To clarify this point, we have added text and modified the discussion section on page 23, beginning on line 504 as “Defective telomere maintenance in MCM10 mutants was not due to a decrease in intrinsic telomerase activity (Fig. 3c, 5d), but is a feature of genome instability caused by the underlying replication defects. In HCT116 MCM10 mutants, we consistently observed telomeres that were shorter than wild type cells, eroded to some extent, and did not recover unless MCM10 expression was restored.”

6) Specifically, it is still not clear what drives the proliferation phenotype in the cells. It can not be telomere length as the super telomerase mutants still grow poorly (figure 5f), as do the RPE-1 cells (figure 4b). Yet, the data indicate that “reversion” occurs when telomeres are short. Why would that happen if the selective pressure is not coming from short telomeres?

The proliferation phenotype is driven by defective DNA replication (Fig. 2). MCM10 is required for origin activation. The most precise and quantitative measure of origin activation is to determine the inter-origin distance (IOD) across the entire genome (Fig. 2g). An increase in IOD is indicative of a decrease in the number of active origins. What seems to be a subtle increase from ~90 to ~120 kb translates into a reduction of active origins to ~25% of wild type (page 10, lines 207 to 208). A decrease in origin firing usually goes hand-in-hand with an increase in replication fork velocity (Rodriguez-Acebes et. al., 2018; DOI:10.1074/jbc/RA118.003740; Zhong et. al., 2013; DOI:10.1083/jcb.201208060) which is modest but statistically significant (Fig. 2h). Fewer active origins translates into fewer active replication forks. To independently confirm that the mutants have fewer active replication forks, we monitored the level of PCNA ubiquitination (Fig. 2j). PCNA can only be ubiquitinated if it is bound to chromatin (Leung et al., 2018; DOI: 10.3390/genes10010010). Ubiquitinated PCNA is reduced in MCM10 mutants. Although this is an indirect readout, it corroborates our claim of fewer replication forks. We have added an analysis of PCNA ubiquitination in RPE-1 mutants (Fig. 4i) that demonstrates a similar phenotype implying that an origin firing defect also decreases the number of active forks in these cells. In addition, in HCT116 mutants the rate of EdU incorporation was diminished (Fig. 2f). These relatively mild aberrations cause genome instability (Supplemental Table 1; Supplemental Fig. 3a,b) and trigger cell death (Fig. 1i, 4c, 7d). Despite the fact that average replication fork stability (when looking at total genomic DNA) is unaffected (Fig. 2i), fork progression through telomeric sequences is significantly impaired (Fig. 6 a-c), which explains the telomere maintenance defect.

With respect to the event that triggers the reversion, we would like to clarify that we believe the reversion is an entirely random event. But telomere shortening (enhanced by the application of a telomerase inhibitor) increases the selection pressure.

We agree that telomere length is not the driver of the proliferation phenotype. For example, we explicitly titled one section of the manuscript “Elevated telomerase activity rescues telomere length but not the inherent replication defect in heterozygous MCM10 mutant cells” (page 17, lines 372 to 373). We have added the following text to clarify our hypothesis: “Our data indicate that the telomere maintenance defect in MCM10 mutant cells is one feature of the underlying defect in maintaining genome stability.” (page 12, line 264 to 266).

7) *It is also surprising that late passage HCT116 MCM10+/- cells have an elevated (but not increased compared to early passage) frequency of signal-free telomere ends (SFEs) (supplementary figure 2g). If “reversion” occurs and telomere length is restored comparable to wt cells, one would expect that the frequency of SFEs to similarly be restored to wt levels. Otherwise, if telomere shortening continues, one would expect elevated SFEs. This apparent contradiction is not addressed.*

It is true that Supplemental Fig. 3e summarizes telomere analysis of late passage cells, however, these were not revertants (and are not described as such). We monitored these cell lines for reversion and did not see any indication of reversion in these particular populations. In our controlled reversion experiment, reversion occurred in 1 out of 5 cultures (page 13, lines 292-293). We quantified SFEs in early (PD <15) and late (PD >60) passage MCM10^{+/-} cell lines and did not detect a significant increase in SFEs in late passage mutants. We have added a

sentence to clarify this data, stating on page 13, lines 273 to 276 that “Thus, although TRF analyses indicated that telomeres in MCM10 mutants eroded over time, there was not sufficient erosion to detect a quantifiable increase in signal free ends on metaphase chromosomes. This observation suggested that late passage MCM10^{+/-} populations maintained minimal telomeres in order to remain viable.”

8) Figures 6a-c explore the inability of super telomerase cells to fully replicate telomeres. However, the resolution of telomere lengths in figure 5c is not sufficient to confirm that WTST3, 8ST5 and 14ST4 have comparable telomere lengths. If anything, WTST3 appears to have slightly shorter telomeres, which could explain its increased ability to fully replicate the telomere within a given window of replication, independent of MCM10 activity.

To directly address this concern, we measured the length of partially or completely replicated telomeres analyzed in Fig. 6c. We have added these data to Supplemental Fig. 5b (included below). These data demonstrate that telomere lengths in wild type and mutant ST cell lines are not significantly different, excluding the possibility that a difference in telomere lengths is the reason for the replication defects reported in Fig. 6a-c. We have added the following text to page 19, lines 410 to 413: “The increase in partially replicated telomeres in MCM10 mutants was not attributable to differences in average telomere length, as measurements of completely and partially replicated telomeres found these to be in an identical size range.”

9) The interpretation of the Mcm10/Mus81 genetic interaction is not satisfactory. While the mutations appear to be additive, the analysis does not clearly indicate a synergistic effect.

We respectfully point out that nowhere in the manuscript do we claim that there is a “synergistic” effect. Based on our data we state that there is “a requirement for Mus81 in promoting cell proliferation and viability in Mcm10-deficient cells, likely through stimulating replication restart in hard-to-replicate regions, including telomeres” (page 21, lines 458 to 460).

This is based not only on our data, but on published literature related to the function of both proteins (reviewed in Falquet & Rass, 2019; DOI:10.3390/genes10030232 and Baxley & Bielinsky, 2017; DOI:10.3390/genes8020073).

Reviewer #3 (Remarks to the Author):

In this manuscript, Baxley et al. report the novel association of compound heterozygous MCM10 loss-of-function (LOF) and hypomorphic variants with a severe fetal phenotype of restrictive cardiomyopathy with lymphoreticular hypoplasia. They propose underlying these clinical features, as well as NK cell deficiency associated with a different set of MCM10 variants described in an accompanying manuscript, is telomere erosion due to a critical role for MCM10 in the restart of stalled replication forks within telomeres with secondary effects on cell proliferation and survival. They present a rigorous body of data analyzing heterozygous MCM10 +/- cells, demonstrating marked effects on telomeres in the HCT116 cancer cell line, with the striking emergence of MCM10 ++ revertants, in hTERT-immortalized RPE-1 cells, particularly in the presence of telomerase inhibition, and in super telomerase HCT116 cells, reflected by accumulation of t complex DNA. They show a role for Mus81 in maintaining telomeres in the context of MCM10 +/- and a lack of a role for replisome proteins Cdc45 and Mcm4. The manuscript makes important contributions to the telomere field although it leaves open the question as why the clinical phenotypes do not overlap more significantly with those associated with mutations in the numerous genes identified to date in the telomere biology disorders. The manuscript would be strengthened by the authors addressing the following points:

We thank the Reviewer for the positive evaluation. We agree that one might expect more overlap with known telomeropathies. The number of cases known-to-date is still limited, and MCM10 is affecting genome replication in addition to telomere maintenance. This might serve as an explanation.

Major points:

1. The cell line data reported here demonstrate MCM10 is genetically haploinsufficient, with striking telomere-related phenotypes and impacts on cell proliferation and survival. GnomAD data, however, suggests that there is not an intolerance of loss-of-function MCM10 variants (pLI 0, o/e 0.84). Additionally, there is an absence of clinical phenotypes in the mother of the family described here or the mother described in the Mace et al., accompanying manuscript, who carried loss-of-function (LOF) variants. Analysis of telomere length (preferably by telomere flow FISH including CD56+ cells) of the parents and offspring without biallelic MCM10 variants would help clarify if MCM10 haploinsufficiency is also observed in primary human cells.

As discussed in our manuscript, the phenotype of the unaffected family members implies that NKD and RCM were not caused by MCM10 haploinsufficiency (page 15, lines 328-330; page 25, lines 559-560). We agree with the Reviewer that this would be a great experiment to better understand MCM10 haploinsufficiency, especially with respect to the specific patient mutations. However, neither tissue nor blood samples from the parents are available to us.

2. The description c.764+5G>A, p.D198GfsTer10 does not make sense to me. First, the coding DNA position 764 does not correspond to the third nucleotide of the D198 codon. Also, p.D198GfsTer10 would not be correct given the predicted impact on RNA splicing. See <https://varnomen.hgvs.org/recommendations/RNA/variant/splicing/> for standard splicing variant nomenclature.

*The Reviewer raises an important point, as the genetic nomenclature for this mutation is not immediately intuitive. Our previous description mischaracterized this variant and its functional consequences, which we have corrected throughout the manuscript. The nucleotide change with respect to the DNA nomenclature is c.764+5G>A (for MCM10 transcript NM_018518.5), located five nucleotides into intron 6. This transition reduces the function of the exon 6 splice donor site but does not completely inactivate it (Supplemental Fig. 1c). Consequently, the donor site of exon 5 can be used and is covalently bound to the acceptor site of exon 7 during pre-mRNA processing. RNA analysis was utilized to define this change as r.593_764del and is predicted to result in the variant protein p.Asp198Glyfs*10. However, bioinformatic analysis found that exon 7 carries a cryptic splice acceptor that would allow expression of full-length MCM10 carrying a deletion of amino acids 198-262. To reflect these details, we have extensively modified the text included in the second paragraph of the Results section (beginning on page 6, line 132), as well as the information in Table 1, Fig. 1c,d, Supplemental Fig. 1b-d).*

3. Pg 23, line 508, I disagree with the statement that cardiomyopathy is a well-recognized feature of dyskeratosis congenita. The 2008 Basel-Vanagaite report highlighted the novelty of cardiac abnormality in patients with DC and proposed this be studied in a larger cohort (which I don't believe has been done). The reviews by Bessler et al., and Savage similarly indicate cardiac abnormalities are rare.

We modified the text on page 25, lines 552 to 553 to “cardiomyopathy has been recognized as a feature of some cases of the telomeropathy dyskeratosis congenita (DC)⁵³⁻⁵⁵, although these cases are rare.”

4. Figures 6a-c demonstrate impaired telomere replication with increased percentage of unreplicated and partially replicated telomeres in the MCM10 +/- ST lines. While the 2D gels clearly show accumulated t-complex DNA, the TRFs analysis in Figure 5c does not show accumulation of short TRFs or increased telomere length heterogeneity. Might this be due to differences in S phase distribution or transit time?

We agree that Fig. 5c does not show an accumulation of short telomeres. However, given the robust overexpression of telomerase (as evidenced by the significantly extended telomeres) and the fact that telomerase preferentially extends short telomeres (Teixeira et. al., 2004: DOI: 10.1016/s0092-8674(04)00334-4), we would not predict an accumulation of short telomeres. Based on the proliferation rates of MCM10 ST mutants (Fig. 5f), we presume that S-phase takes longer than in wild type cells. However, it is unclear whether increased total duration of S-phase corresponds with an increased opportunity to extend telomeres in these mutants. Our new analysis measuring partially or completely replicated telomeres by DNA combing (Supplemental Fig. 5b; included above in response to Reviewer 2, additional concern #8) shows similar

telomere lengths in wild type and mutant cells, suggesting that a longer S-phase does not generate significantly longer telomeres.

5. If MCM10 contributes to telomere length maintenance by stimulating replication restart within duplex telomeric tracts, then the requirement or impact of MCM10 deficiency might differ in cells with long versus short telomeres with greater impact on those with longer telomeres. Can attrition rates be determined in the various cell lines, and possibly in the ST lines following telomerase inhibition?

To analyze telomere attrition when telomerase activity was inhibited, we cultured cells for 50 to 80 population doublings (PDs; Fig. 4n and 5a), which required 8 to 12 weeks in culture. We estimate the attrition rates for both wild type and MCM10 mutant cells to be ~20 to 25 bp per PD. These values are consistent with erosion due to the end-replication problem and have been added to the text on page 17, line 379. However, for RPE-1 wild type cells treated with the same concentration of telomerase inhibitor we estimated the attrition rates to be ~7 bp per PD, and ~16 bp per PD in the MCM10 mutant and have been added this information to the text on page 16, lines 360-362. These data suggest that the concentration of inhibitor used was insufficient to completely block telomerase activity in RPE-1 cells. We have added the following text to page 17, lines 380 to 382 as an explanation for these differences: “Thus, telomerase inhibition was dominant to the effect of Mcm10 deficiency on telomere length regulation, as telomere attrition in MCM10 mutants was not exacerbated when telomerase was completely inhibited”.

To accurately determine attrition rates in ST cells lines, we would need to identify inhibitor concentrations sufficient to completely block telomerase in each cell line, as these cells have significantly increased telomerase activity. Given the modest rate of erosion in non-ST HCT116 cells lines, and that telomeres in ST cells are at least 4 times longer than in non-ST cells (~25 to 30 kb versus 4 to 6 kb, respectively), we estimate the amount of time required to complete these initial experiments to be a minimum of 4 times longer than for non-ST lines (at least 8 weeks), or approximately a minimum of 7 months. Therefore, accurately defining the rate of attrition in ST lines is beyond the scope of the current manuscript.

The Reviewer raises an interesting point regarding the impact of MCM10 deficiency and telomere length in non-ST and ST cell lines. Ideally, to address this we would perform telomere combing experiments in non-ST cell lines, but this is technically impossible due to the length of telomeres in non-ST HCT116 cell lines. However, we reasoned that if exceptionally long telomeres are more deleterious than short telomeres, then ST cell lines would exhibit decreased clonogenic survival and growth. Therefore, we expanded our previous colony formation analyses to include multiple time points (Fig. 5g, Supplemental Fig. 5a). These data show that clonogenic survival in wild type cells is slightly decreased for ST cells, suggesting that long telomeres have a negative impact. However, no difference was observed when comparing non-ST with ST MCM10-deficient cell lines. We propose that although long telomeres may reduce survival in MCM10-deficient cells, telomerase overexpression also rescues cell death due to critically short telomeres. This balancing effect results in no discernable change in net survival.

6. Pg 13, lines 292-3, the statement that Cdc45 levels remained normal in MCM10^{+/-} cells lines is not supported by the data which is variable with levels as low as 0.6X WT (Supplemental fig. 3).

The Reviewer is correct. We modified the text on page 14, lines 311 to 314 as follows: "We confirmed that Cdc45 or Mcm4 levels were not consistently reduced in MCM10 mutants, implying that the mutant phenotypes in these cells are not attributable to reduced expression of either factor."

Minor points:

7. The word mutation(s) should be replaced by variant(s) (see Richards et al, Standards and guidelines for the interpretation of sequence variants: a joint consensus recommendation of the American College of Medical Genetics and Genomics and the Association for Molecular Pathology, Genet Med 2015).

Thank you. We modified the text to reflect this change.

8. Not all of the research subjects were patients e.g., parents and unaffected siblings.

Thank you. We modified the text to reflect this change.

9. Pg 3, line 60, CM should be RCM.

The Reviewer is correct, and we made this change.

10. Pg 5, lines 104-106, given a strong telomere phenotype was observed MCM10 ^{+/-}, it suggests that MUS81 is unable to prevent fork collapse.

We agree, but not having MUS81 clearly makes matters worse. We modified the text on page 5, lines 105 to 107 as follows: "Although likely not the only nuclease involved, our data demonstrate that MCM10 mutants utilize Mus81 to process stalled replication forks and improve cell survival."

11. Pg 6, line 125 and Pg 8, line 163, c.236delG is not a nonsense variant.

The Reviewer is correct, the variant should be referred to as a frameshift variant, not a nonsense variant. We have modified the text to reflect this change.

12. Pg 10, the description of numbers of novel karyotypes but reference to a table that lists specific chromosomal aberrations (not karyotypes) is confusing.

We understand the Reviewer's confusion here. We modified the text such that Supplemental Table 1 is only referred to in the context of common fragile sites (CFSs), which is specifically what the table is representing.

13. Pg 11, line 23, highlight that the other cells were those with exon 14 targeted, as it is not stated. Indeed, it might be good to state early that for most experiments clones obtained after exon 14 targeting were analyzed.

We believe the Reviewer would like clarification in the paragraph on beginning of page 11, line 248 when exon 3 or exon 14 targeted MCM10 mutant cell lines were analyzed. We modified the text on page 12, lines 248 to 251 to address this comment stating, “We confirmed this phenotype in early passage exon 3 MCM10^{+/-} HCT116 cells, which showed significantly eroded telomeres (Supplemental Fig. 3c). Subsequent analyses focused on exon 14 MCM10 mutant cell lines.”

14. Pg 13, line 294, Fig. 1d should be Fig. 1f.

Thank you. We corrected that mistake.

15. Pg 20, line 437, the RPE-1 cells are TERT immortalized, correct? This needs to be stated.

This detail was stated on page 5, line 96, but has been added to page 22, lines 480 to 481 that the Reviewer is referring to for emphasis.

16. Pg 23, line 520, if amino acids 198-255 are deleted in the splice site variant (pg 7, line 138), then the number of deleted amino acids should be 258 not 255.

This text has been changed to reflect changes described in our response to major point #2 above.

17. Pg 25, line 553, spell-out GM-MDT.

We spelled out GM-MDT.

18. Supplemental Figure 1b legend, it should be explicitly stated that sequencing is of cDNA, presumably cloned for the father.

The Reviewer is correct, and we made that change.

19. How was the structural model of human Mcm10-ID in Fig. 1c generated or is it the Xenopus structure with the position of the cognate R426C residue and those that would be deleted if human exon 6 was skipped shown?

The following was stated in the figure legend for Fig. 1c, “Structural model of the human Mcm10-ID with a bound single-stranded DNA, based on Xenopus laevis Mcm10-ID.”

20. Add kB above ladder numbers in Fig. 5c left and right telomere Southern blots.

The Reviewer is correct, and we made this change.

21. DC is now more frequently used than DKC to abbreviate dyskeratosis congenita to limit assumed reference to DKC1.

The Reviewer raises a good point. We have abbreviated dyskeratosis congenita as DC rather than DKC.

REVIEWERS' COMMENTS

Reviewer #1 (Remarks to the Author):

The authors are thanked for their careful and thorough revisions, which have addressed all my previous comments. All modifications, including those in response to the other reviewers, further strengthen and clarify the findings presented. As articulated previously it is felt this study presents a timely and important advance in the field, as it illuminates previously unappreciated cellular phenotypes conferred by limiting dosage of MCM10.

In the revised text, there were three minor suggestions:

line 260-261. For readers who might not know what these other sources of stress-induced senescence might be, the authors could include this recent reference, which summarizes them in Table 1 within that citation: <https://doi.org/10.1016/j.cell.2019.10.005>

line 274. Authors could change "not sufficient" to "insufficient".

Overall nomenclature. According to international guidelines, human proteins should be in all caps, with no italics. The authors may wish to check this and modify their text accordingly.

Reviewer #2 (Remarks to the Author):

I find that the authors did not address several of my key issues that I outlined in my suggestions.

- The telomere length data does not have the necessary quality to support the conclusion.
- The differences between the cell types are striking and remain unexplained. The authors side with the phenotype seen in cancer cells over the lack of phenotypes seen in fibroblasts. This seems quite risky considering the nature of the mutation in patients?
- The presence of the LoxP site remains a major concern in the interpretation of the putative wild-type allele.
- The paper now ends on "Future analysis of different human cell lineages and/or separation-of-function MCM10 mutations would be required to demonstrate that MCM10-dependent telomere maintenance defects are causative in these human diseases." Without establishing MCM10 mutations as causative for a "telomeropathy" that impact of the study remains unclear.

Reviewer #3 (Remarks to the Author):

In this revised manuscript, Baxley et al, have adequately addressed my prior concerns. The recent discovery of MCM10-associated NK cell deficiency and, in this manuscript, restrictive cardiomyopathy with lymphoreticular hypoplasia, warrants a greater understanding of how MCM10-deficiency impacts cellular phenotypes. While this current work focuses on analyses of HCT116 and RPE-1 cell lines (not relevant to the selectively affected tissues in the affected humans), it provides convincing data to implicate MCM10 (but, notably, not MCM4 or Cdc45) in telomere replication and length maintenance. This work should be of broad interest, including those in the telomere, DNA replication, immunology and molecular cardiology fields. There remain a few issues that should be addressed.

1. The text on the bottom of page 12 – page 13 indicates that independent MCM10^{+/-} subpopulations @ >75 PDs were analyzed (as shown in Figures 3d and 3f) but the signal free end and fragile telomere analyses (Supplemental Fig. 3e) appears to have been performed with bulk population nonrevertant cells (as stated in the rebuttal to Reviewer 2, item 7). Given half of the subpopulations had reverted MCM10, which cells were examined is crucial for the interpretation of Supplemental Figure 3e. This needs to be clarified in the main text.
2. The statistical analysis of Supplemental Figure 5a is not clear. The text suggests the

comparisons are between non-ST and ST for wild type and the MCM+/-10 lines but the analyses appear to be between non-ST WT and each of the others.

3. The discussion on page 7 related to the functional score of exon 6 variant splice donor being great than that of exon 5 but functional score of the cryptic splice acceptor exon 7 being similar to that of the canonical exon 7 acceptor seems to emphasize the differences in the scores for the former and ignore the differences in the scores for the latter comparisons.

Additional minor points

1. The key for Supplemental Figure 1c is reversed, is it not? The top (light gray) should represent the splice donor and the bottom (dark gray) should represent the splice acceptor.

2. Lines 175-176 p.F583X and p.R582X.

3. Line 240 – not all of the aberrations are translocations. Perhaps “events” would be a better word.

4. Line 244 – same as above. Perhaps “chromosomal aberrations” rather than “translocation hotspots” would be better.

5. Supplemental table 1 - bottom of table – “Overlap with of MCM10+/- aberrations with CFSs – omit “with” and correct to aberrations.

6. The chromosomal aberration descriptions in Supplemental Fig. 3b should correspond with those in Supplemental Table 1 (e.g., t(5;8) not transl(5;8)).

7. If a statement is to be made regarding the relative number of signal free ends in early and late passage MCM10+/- clones, then the data should be presented side by side and statistical analysis performed.

Reviewer comments are listed as black text. Our responses are listed below each comment in *blue italicized text*. Our references to page and line number refer to the revised version of our manuscript.

Reviewer #1 (Remarks to the Author):

The authors are thanked for their careful and thorough revisions, which have addressed all my previous comments. All modifications, including those in response to the other reviewers, further strengthen and clarify the findings presented. As articulated previously it is felt this study presents a timely and important advance in the field, as it illuminates previously unappreciated cellular phenotypes conferred by limiting dosage of MCM10.

In the revised text, there were three minor suggestions:

1) Line 260-261. For readers who might not know what these other sources of stress-induced senescence might be, the authors could include this recent reference, which summarizes them in Table 1 within that citation: <https://doi.org/10.1016/j.cell.2019.10.005>

We thank the reviewer for this suggestion and have included this reference.

2) Line 274. Authors could change "not sufficient" to "insufficient".

We have made this change.

3) Overall nomenclature. According to international guidelines, human proteins should be in all caps, with no italics. The authors may wish to check this and modify their text accordingly.

We thank the reviewer for this suggestion and have modified the text accordingly.

Reviewer #2 (Remarks to the Author):

I find that the authors did not address several of my key issues that I outlined in my suggestions.

1) The telomere length data does not have the necessary quality to support the conclusion.

We respectfully disagree. Our telomere length analyses throughout the paper (Figure 3a, 3d, 3g, 5b-d, 6a-b, 8e, and Supplementary Figures 3c, 3h) consistently demonstrate that multiple MCM10 variants/mutations cause MCM10 deficiency that corresponds with shorter telomeres in two distinct cell types.

2) The differences between the cell types are striking and remain unexplained. The authors side with the phenotype seen in cancer cells over the lack of phenotypes seen in fibroblasts. This seems quite risky considering the nature of the mutation in patients?

We respectfully point out that fibroblasts were not used in our study. We utilized HCT116 colorectal cancer cells and hTERT immortalized, non-transformed retinal pigment epithelial (RPE-1) cells. As discussed in the manuscript and our previous responses to reviewer comments, both cell types displayed key similarities, including lower MCM10 expression levels (Figure 1e,

3h, 4a, 4i, 6e, 8a), slower proliferation rate (Figure 1f, 3i, 4b, 6f, 8b and Supplementary Figure 5c), increased cell death (Figure 1i, 4c, 8d), a reduced number of active replication forks (Figure 2g, 2j, 4i) and reduced telomere length (Figure 3a, 3d, 3g, 5b-d, 6a-b, 8e, and Supplementary Figures 3c, 3h). In fact, telomere analyses were performed for RPE-1 mutants that specifically modeled MCM10 patient variants (Figure 5d). In addition to these similarities, we did observe more severe phenotypes in HCT116 cells in comparison to the RPE-1 cells. Based on these observations, we propose a model in which different cell lineages have distinct and inherent thresholds for MCM10 expression, which would also serve to explain why the patient-associated MCM10 variants caused tissue-specific pathologies.

3) The presence of the LoxP site remains a major concern in the interpretation of the putative wild-type allele.

Variations of the Cre-Lox system have been used for many years in a wide variety of model systems and human cell lines. Its usage necessitates the integration of two loxP sites flanking the region to be deleted, which is reduced to a loxP “scar” following recombination. The reviewer’s comments seem to question the validity of studies utilizing any loxP containing allele. In our study, we generated heterozygous MCM10 HCT116 mutant cell lines using two distinct gene targeting strategies (CRISPR-Cas9 or rAAV) to disrupt two different exons (exon 3 or exon 14; Figure 1a). Our data clearly demonstrate that the level of full-length MCM10 expression is similar in heterozygous cell lines regardless of targeting strategy used or whether the remaining functional allele does (exon 14) or does not (exon 3) carry a loxP site downstream of exon 14 (Figure 1e).

4) The paper now ends on “Future analysis of different human cell lineages and/or separation-of-function MCM10 mutations would be required to demonstrate that MCM10-dependent telomere maintenance defects are causative in these human diseases.” Without establishing MCM10 mutations as causative for a "telomeropathy" that impact of the study remains unclear.

Our modeling of MCM10 patient variants demonstrates that MCM10 deficiency causes reduced telomere length as one feature of genome instability arising from underlying replication defects and is consistent with our proposal that these pathologies may be classified as telomeropathies. What remains unclear is whether the telomere maintenance defects are uniquely or primarily causative of these diseases, or whether genome instability at non-telomeric loci is the major driver of pathology. Because of this, we carefully phrased this sentence to emphasize that future analysis would be required to definitively show this.

Reviewer #3 (Remarks to the Author):

In this revised manuscript, Baxley et al, have adequately addressed my prior concerns. The recent discovery of MCM10-associated NK cell deficiency and, in this manuscript, restrictive cardiomyopathy with lymphoreticular hypoplasia, warrants a greater understanding of how MCM10-deficiency impacts cellular phenotypes. While this current work focuses on analyses of HCT116 and RPE-1 cell lines (not relevant to the selectively affected tissues in the affected humans), it provides convincing data to implicate MCM10 (but, notably, not MCM4 or Cdc45) in telomere replication and length maintenance. This work should be of broad interest, including those in the telomere, DNA replication, immunology and molecular cardiology fields. There

remain a few issues that should be addressed.

1) The text on the bottom of page 12 – page 13 indicates that independent MCM10^{+/-} subpopulations @ >75 PDs were analyzed (as shown in Figures 3d and 3f) but the signal free end and fragile telomere analyses (Supplemental Fig. 3e) appears to have been performed with bulk population nonrevertant cells (as stated in the rebuttal to Reviewer 2, item 7). Given half of the subpopulations had reverted MCM10, which cells were examined is crucial for the interpretation of Supplemental Figure 3e. This needs to be clarified in the main text.

The reviewer raises an important point. We have added text to clarify our description (as listed below) as well as added the PCR genotyping data for late passage populations analyzed by t-FISH (Supplemental Fig 3f).

- 1) *“Consistent with these data, the frequency of signal free ends in two additional independent MCM10^{+/-} populations remained elevated in MCM10^{+/-} cells” (page 12-13, line 273-274).*
- 2) *“Both late passage populations analyzed by t-FISH showed the expected mutant genotype (Supplemental Fig 3f)” (page 13, line 280-281).*

2) The statistical analysis of Supplemental Figure 5a is not clear. The text suggests the comparisons are between non-ST and ST for wild type and the MCM^{+/-}-10 lines but the analyses appear to be between non-ST WT and each of the others.

The reviewer is correct that statistics were determined for the comparison of non-ST wild type and each other cell line. Differences between non-ST and ST MCM10 mutant cell lines were not statistically significant. We have updated the text (page 18, lines 413-419), figure and figure legend to indicate these results.

3) The discussion on page 7 related to the functional score of exon 6 variant splice donor being greater than that of exon 5 but functional score of the cryptic splice acceptor exon 7 being similar to that of the canonical exon 7 acceptor seems to emphasize the differences in the scores for the former and ignore the differences in the scores for the latter comparisons.

The reviewer raises a valid point. We have modified the text to more clearly describe the relative functional score of the exon 7 cryptic splice acceptor as follows (page 7, lines 143-145):
“Notably, our analysis also identified a cryptic splice acceptor in exon 7 with a reduced functional score in comparison to the canonical exon 7 acceptor, although higher than the exon 5 acceptor.”

Additional minor points

1) The key for Supplemental Figure 1c is reversed, is it not? The top (light gray) should represent the splice donor and the bottom (dark gray) should represent the splice acceptor.

The reviewer is correct. We have corrected the figure and figure legend.

2) Lines 175-176 p.F583X and p.R582X.

The MCM10 protein mutation designations on these lines (176-177) have been corrected.

3) Line 240 – not all of the aberrations are translocations. Perhaps “events” would be a better word.

The reviewer is correct. We have modified the text as suggested.

4) Line 244 – same as above. Perhaps “chromosomal aberrations” rather than “translocation hotspots” would be better.

The reviewer is correct. We have modified the text as suggested.

5) Supplemental table 1 - bottom of table – “Overlap with of MCM10[±] aberrations with CFSs – omit “with” and correct to aberrations.

Supplemental table 1 has been corrected as suggested by the reviewer.

6) The chromosomal aberration descriptions in Supplemental Fig. 3b should correspond with those in Supplemental Table 1 (e.g., t(5;8) not transl(5;8)).

The aberration descriptions in Supplemental Fig. 3b have been corrected to match Supplemental Table 1.

7) If a statement is to be made regarding the relative number of signal free ends in early and late passage MCM10[±] clones, then the data should be presented side by side and statistical analysis performed.

This comment is in reference to the text on page 12/13, lines 274-276, of the previously revised manuscript as follows: “Consistent with these data, the frequency of signal free ends in two additional independent MCM10[±] populations remained elevated in MCM10[±] cells, but was not increased in comparison to early passage mutants (Supplemental Fig. 3e, Fig. 3b).” To address the reviewer’s comment, we have removed the text comparing early and late passage mutants, corresponding to the text on page 12-13, line 273-274 of the current manuscript.